# Phosphate steering by Flap Endonuclease 1 promotes 5′-flap specificity and incision to prevent genome instability

Susan E. Tsutakawa[1,*], Mark J. Thompson[2,*], Andrew S. Arvai[3,*], Alexander J. Neil[4,*], Steven J. Shaw[2], Sana I. Algasaier[2], Jane C. Kim[4], L. David Finger[2], Emma Jardine[2], Victoria J.B. Gotham[2], Altaf H. Sarker[5], Mai Z. Her[1], Fahad Rashid[6], Samir M. Hamdan[6], Sergei M. Mirkin[4], Jane A. Grasby[2] & John A. Tainer[1,7]

DNA replication and repair enzyme Flap Endonuclease 1 (FEN1) is vital for genome integrity, and FEN1 mutations arise in multiple cancers. FEN1 precisely cleaves single-stranded (ss) 5′-flaps one nucleotide into duplex (ds) DNA. Yet, how FEN1 selects for but does not incise the ss 5′-flap was enigmatic. Here we combine crystallographic, biochemical and genetic analyses to show that two dsDNA binding sites set the 5′polarity and to reveal unexpected control of the DNA phosphodiester backbone by electrostatic interactions. Via 'phosphate steering', basic residues energetically steer an inverted ss 5′-flap through a gateway over FEN1's active site and shift dsDNA for catalysis. Mutations of these residues cause an 18,000-fold reduction in catalytic rate *in vitro* and large-scale trinucleotide $(GAA)_n$ repeat expansions *in vivo*, implying failed phosphate-steering promotes an unanticipated lagging-strand template-switch mechanism during replication. Thus, phosphate steering is an unappreciated FEN1 function that enforces 5′-flap specificity and catalysis, preventing genomic instability.

[1] Molecular Biophysics and Integrated Bioimaging, Lawrence Berkeley National Laboratory, Berkeley, California 94720, USA. [2] Centre for Chemical Biology, Sheffield Institute for Nucleic Acids (SInFoNiA), Department of Chemistry, University of Sheffield, Sheffield S3 7HF, UK. [3] Department of Molecular Biology, The Scripps Research Institute, La Jolla, California 92037, USA. [4] Department of Biology, Tufts University, Medford, Massachusetts 02155, USA. [5] Biological Systems and Engineering, Lawrence Berkeley National Laboratory, Berkeley, California 94720, USA. [6] Division of Biological and Environmental Sciences and Engineering, King Abdullah University of Science and Technology, Thuwal 23955, Saudi Arabia. [7] Department of Molecular and Cellular Oncology, The University of Texas M.D. Anderson Cancer Center, Houston, Texas 77030, USA. * These authors contributed equally to this work. Correspondence and requests for materials should be addressed to S.M.M. (email: sergei.mirkin@tufts.edu) or to J.A.G. (email: j.a.grasby@sheffield.ac.uk) or to J.A.T. (email: jtainer@mdanderson.org).

The structure-specific nuclease, flap endonuclease-1 (FEN1) plays a vital role in maintaining genome integrity by precisely processing intermediates of Okazaki fragment maturation, long-patch excision repair, telomere maintenance, and stalled replication forks. During DNA replication and repair, strand displacement synthesis produces single-stranded (ss) 5′-flaps, at junctions in double-stranded (ds) DNA. During replication in humans, FEN1 removes ∼50 million Okazaki fragment 5′-flaps with remarkable efficiency and selectivity to maintain genome integrity[1–3]. Consequently, FEN1 deletion is embryonically lethal in mammals[4], and functional mutations can lead to cancer[5]. FEN1 also safeguards against DNA instability responsible for trinucleotide repeat expansion diseases[6]. As FEN1 is overexpressed in many cancer types[7,8], it is an oncological therapy target[9,10].

Precise FEN1 incision site selection is central to DNA replication fidelity and repair. FEN1 preferentially binds to double flap substrates with a one nt 3′-flap and any length of 5′-flap, including zero. It catalyses a single hydrolytic incision one nucleotide (nt) into dsDNA (Fig. 1a) to yield nicked DNA ready for direct ligation[11,12]. Thus, FEN1 acts on dsDNA as both an endonuclease (with 5′-flap) and an exonuclease (without 5′-flap). Recent single molecule experiments show that FEN1 binds both ideal and non-ideal substrates but decisively incises only its true substrate[13]. In contrast to homologs in bacteriophage[14–16] and some eubacteria[17], eukaryotic FEN1s do not hydrolyse within 5′-flap ssDNA.

However, key features of FEN1 substrate selection remain unclear. FEN1 must efficiently remove 5′-flaps at discontinuous ss-dsDNA junctions yet avoid genome-threatening action on continuous ss–ds junctions, such as ss gaps or Holliday junctions. Paradoxically, other FEN1 5′-nuclease superfamily members[3] are specific for continuous DNA junctions: namely, ERCC5/XPG (nucleotide excision repair), which acts on continuous ss-ds bubble-like structures; and GEN1 (Holliday junction resolution), which processes four-way junctions. Structures determined with DNA of eukaryotic superfamily members lack ss-ds junction substrate with 5′-ssDNA or the attacking water molecule leaving cardinal questions unanswered[18–22]. For example, structures of FEN1 and Exo1 go from substrate duplex DNA with the scissile phosphodiester far from the catalytic metals to an unpaired terminal nt in the product; is the unpairing occurring before or after incision?

Models of FEN1 specificity must address how ss–ds junctions are recognized and how 5′-flaps, as opposed to continuous ssDNA are recognized. There are threading and kinking models. To exclude continuous DNAs, 5′-flaps may thread through a 'tunnel'[21,23–25] formed by two superfamily-conserved helices flanking the active site, known as the 'helical gateway,' topped by a 'helical cap' (Fig. 1b). Due to cap and gateway disorder in DNA-free FEN1, they are thought disordered during threading and to undergo a disorder-to-order transition on 3′-flap binding[21,24,26]. In this threading model, however, ssDNA passes through a tunnel without an energy source and directly over the active site, risking non-specific incision. These issues prompted an alternative clamping model where the ss flap kinks away from the active site[11,20] (Fig. 1b). Whereas these models explain selection against continuous DNA junctions, FEN1 exonuclease activity does not require a 5′-flap. Furthermore, how FEN1 prevents off target incisions and moves the dsDNA junction onto the metal ions are not explained by these models.

Here crystallographic analyses uncover an unprecedented electrostatic steering of an inverted 5′-flap through the human FEN1 (hFEN1) helical gateway. Gateway and cap positively-charged side chains are positioned to 'steer' the phosphodiester backbone across the active site, energetically promoting threading and preventing nonspecific hydrolysis within the 5′-flap. Mutational analysis of these positively charged 'steering' residues revealed an added role of phosphate steering in moving dsDNA towards the catalytic metal ions for reaction. Moreover, phosphate steering mutations efficiently blocked Rad27 (S. cerevisae homolog of hFEN1) function, causing a compromised response to DNA damaging agents and dramatically increased expandable repeat instability.

## Results

**FEN1 selects for 5′-flaps by steering flap through a gateway.** To obtain structures of hFEN1 with a ss 5′-flap substrate for insight into ss 5′-flap selection, we crystallized three hFEN1 active site mutants D86N, R100A and D233N with a double-flap (DF) substrate and with $Sm^{3+}$ (Fig. 1 and Supplementary Figs 1 and 2A)[27]. $Mg^{2+}$ is the physiological cofactor. D86N, R100A and D233N mutations slow the hFEN1 catalysed reaction rate by factors of 530, 7,900 and 16 respectively (Supplementary Fig. 2B). The DF substrates in the crystal structures had a ss 5′-flap (4–5 nt) and a 1 nt 3′-flap, termed S4,1 or S5,1 (Supplementary Fig. 1). The DNA-enzyme complex structures for hFEN1-D86N, hFEN1-R100A, and hFEN1-D233N were determined to 2.8, 2.65, and 2.1 Å resolution, respectively (Figs 1c,d and 2, Supplementary Fig. 2 and Table 1). In all cases, the overall protein resembled wild-type (wt) hFEN1 (with product DNA, PDB code 3Q8K)[21], with root mean square deviation (RMSD) values of 0.26 for hFEN1-R100A, 0.22 for hFEN1-D233N, and 0.42 for hFEN1-D86N.

These structures show that FEN1 interacts primarily (88% by PISA interface analysis[28]) with two regions of ∼100° bent dsDNA supporting prior observations[21], rather than to the ss 5′-flap in these structures (Figs 1c,d and 2, Supplementary Movie 2). FEN1 binding to dsDNA is mediated by four regions: (1) a hydrophobic wedge (composed of helix 2 and helix 2–3 loop) and β pin (formed between β strands 8 and 9) sandwich upstream and downstream dsDNA portions at the bending point of the two-way junction with Tyr40 packing at the ss/dsDNA junction; (2) a C-terminal helix-hairpin-helix motif binds upstream dsDNA and the one nt 3′-flap and is absent from superfamily-related members hEXO1 (ref. 20) and bacteriophage 5′-nuclease structures[29]; (3) the helix-2turn-helix (H2TH) motif with bound $K^+$ ion and positive side chains bind downstream dsDNA; and (4) the two-metal ion active site near the 5′-flap strand. Much of the interaction (43% by PISA analysis) is to the strand complementary to the flap strands, reinforcing dsDNA specificity.

The dsDNA binding sites on either side of the active site, the $K+$ and the hydrophobic wedge, are spaced one helical turn apart (Supplementary Movie 2). Their spacing enforces the specificity for helical dsDNA and places the 5′-flap in the active site, selecting against unstructured ssDNA or 3′-flaps that would require a narrower spacing. Additionally, the minor-groove phosphate backbone is recognized by superfamily-conserved Arg70 and Arg192 pair spaced 14 Å apart (Fig. 2, Supplementary Movie 1). Unique to FEN1, cap positive side chains (Lys125, Lys128, Arg129) interact with the template strand at the ss/dsDNA junction (Supplementary Fig. 3, Supplementary Movie 1). Lys128 and Arg47 pack against each other, linking the 3′-flap pocket to the gateway helices. The active site consists of seven superfamily-conserved metal-coordinating carboxylate residues plus invariant Lys93 and Arg100 from gateway helix 4 and Gly2 at the processed N-terminus (Figs 2 and 3c, Supplementary Movie 4). An ordered gateway and cap formed by helices 2, 4 and 5 are observed above the active site in these three structures. Helix 2 Tyr40 forms part of the hydrophobic wedge and packs against the duplex DNA at the bend.

In the hFEN1-D86N and hFEN1-R100A structures, the ssDNA (5′-flap) region of the substrate threaded through the tunnel formed by the gateway/cap (Figs 1d and 2a, and Supplementary Fig. 2A and Supplementary Movie 1). This observation explains how FEN1 excludes continuous DNA like Holliday junctions and DNA bubbles. The third independent hFEN1-D233N crystal structure captures two cleaved nts from the 5′-flap bound on the other side of the tunnel from the dsDNA, consistent with threading (Fig. 3a and Supplementary Fig. 2A). Together, these three distinct structures support the threading model to validate substrates have a ss 5′-flap.

**Phosphate steering inverts the ss phosphodiester backbone.** In both threaded substrate structures, the ss 5′-flap phosphodiester backbone is 'inverted' between the +1 and +2 positions, with the +2 and +3 phosphates facing away from the active site metals and the DNA bases facing the metals (Fig. 1b,d and Supplementary Movie 1). (We denote the plus and minus positions relative to the scissile phosphate). This inversion would place the flap phosphodiesters away from the catalytic metals and thereby logically reduce inadvertent incision within the ssDNA. In both structures, the inverted +1 phosphodiester is directly between the gateway helices with the bases on either side of the

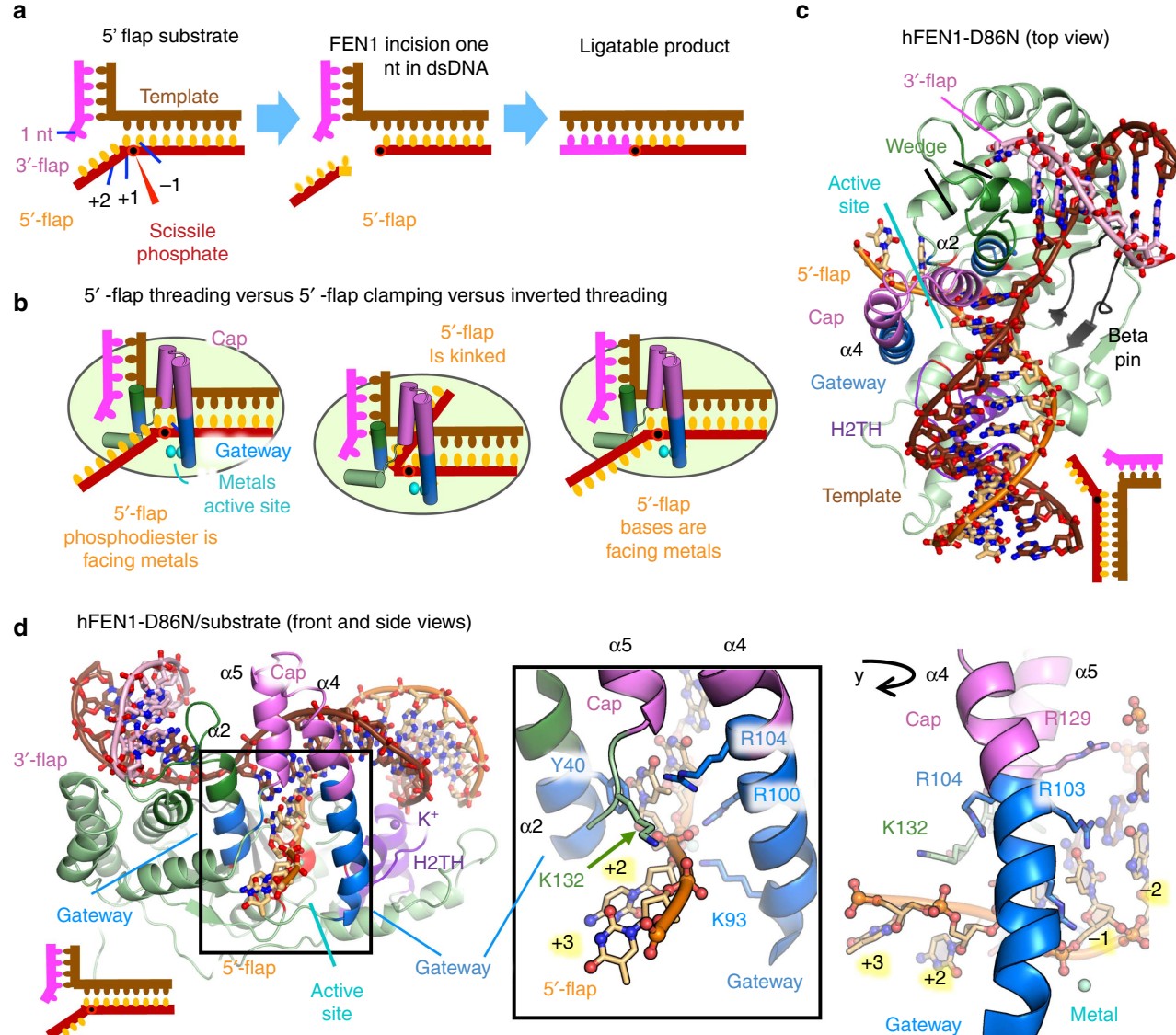

**Figure 1 | Specificity and inverted threading of ss5′-flap in hFEN1 D86N substrate structure.** (**a**) Schematic FEN1 incision on an optimal double-flap substrate, incising 1 nt into dsDNA to ensure a ligatable product. (**b**) Proposed models for ssDNA selection. (**c**) Top view of hFEN1-D86N crystal structure showing extensive interaction to dsDNA arms of 5′ flap substrate. The 5′-flap substrate is composed of three strands; the 5′-flap strand (orange), the template strand (brown), and the 3′-flap strand (pink). Functionally critical regions in FEN1 include the gateway (blue) and the cap (violet) for selecting substrates with ss-5′-flaps, the hydrophobic wedge between the 3′-flap binding site and the gateway/cap (dark green), the K+ ion and H2TH (purple) that interacts with the downstream DNA, the beta pin (grey) that locks in the DNA at the bend. Relative DNA orientation shown in schematic on lower right. (**d**) Front and side views of hFEN1-D86N crystal structure showing helical gateway and cap architecture position positively-charged residues to steer ss 5′-flaps through a protecting gateway in an inverted orientation across the active site. Relative DNA orientation is shown in schematic. The inverted 5′-flap ssDNA is threaded between gateway helices (blue) and under the helical cap (violet). The inverted threading reveals charged interactions to basic sidechains in the cap and van der Waals interactions to ssDNA. See also Supplementary Figs 1–3; Table 1, and Supplementary Movies 1 and 2.

gateway. In the hFEN1-D86N structure, two basic residues of the gateway/cap, Arg104 and Lys132, were within 4–7 Å of the +1, +2 and +3 phosphodiester. These residues are positioned to energetically promote threading and an inverted orientation. They are conserved in FEN1 and semi-conserved across the 5′-nuclease superfamily and shown important for incision activity

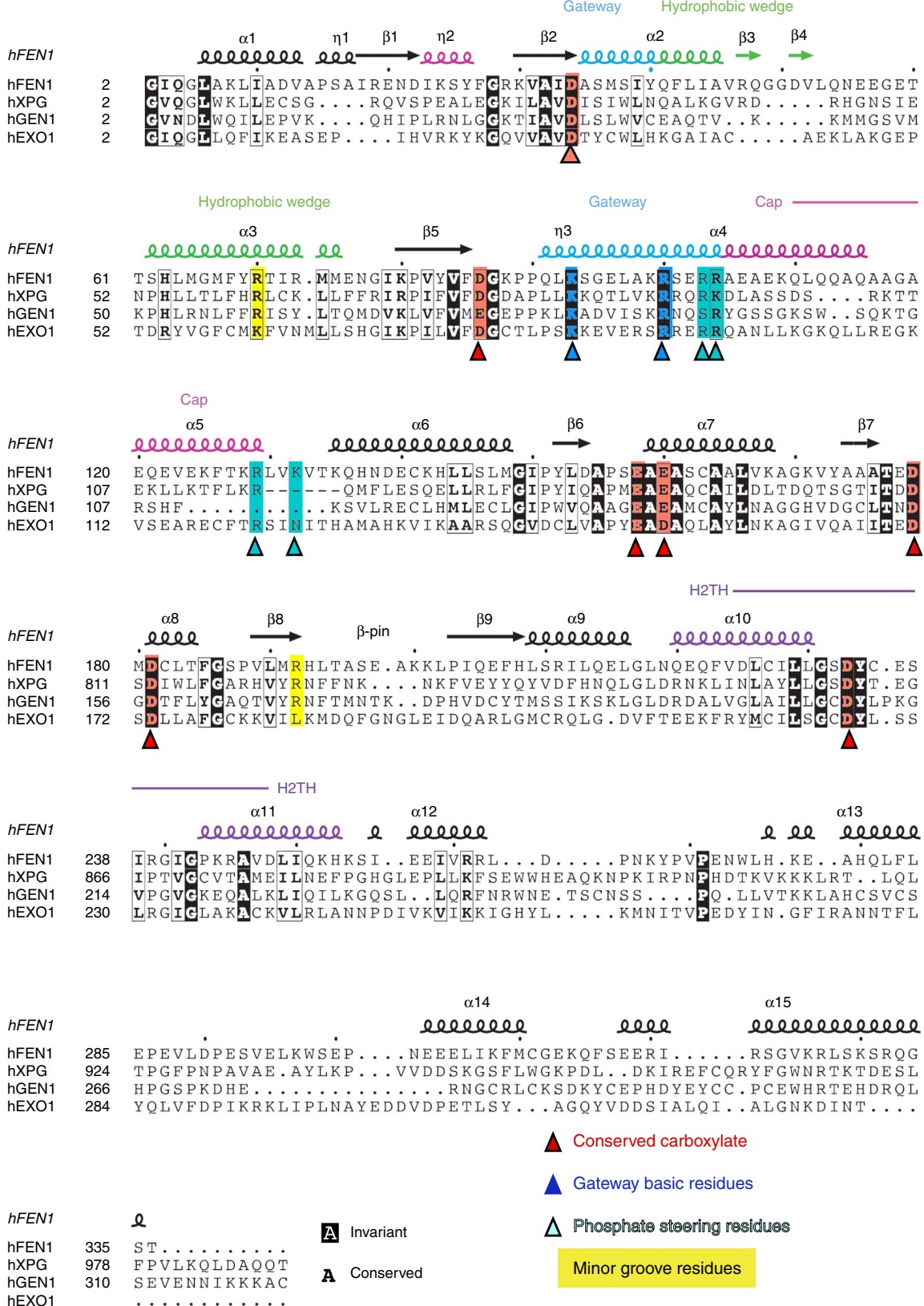

**Figure 2 | FEN1 superfamily sequence and secondary structure alignment.** Map of FEN1 secondary structure (PDB code 3q8k), structural elements, and mutants to a sequence alignment of FEN superfamily human members. XPG residues 117–766 were removed (dash) to facilitate alignment.

**Table 1 | X-ray data collection and refinement statistics (molecular replacement).**

|  | hFEN1-D86N | hFEN1-R100A | hFEN1-D233N |
|---|---|---|---|
| *Data collection* |  |  |  |
| Space group | P 31 2 1 | P 31 2 1 | P 31 2 1 |
| Cell dimensions |  |  |  |
| $a, b, c$ (Å) | 105.8 105.8 100.7 | 105.2 105.25 104.1 | 105.2 105.2 104.5 |
| $\alpha, \beta, \gamma$ (°) | 90 90 120 | 90 90 120 | 90 90 120 |
| Resolution (Å) | 18.2–2.8 (2.9–2.8) | 37.0–2.6 (2.7–2.6) | 34.5–2.1 (2.2–2.1) |
| $R_{meas}$ | 0.08 (1.0) | 0.07 (0.70) | 0.09 (0.65) |
| $I/\sigma I$ | 22.7 (1.95) | 12.7 (2.9) | 73.9 (5.00) |
| Completeness (%) | 1.00 | 0.98 | 0.99 |
| Redundancy | 8.6 (8.5) | 5.2 (5.4) | 13.8 (10.4) |
| *Refinement* |  |  |  |
| Resolution (Å) | 18.2–2.8 | 37.0–2.6 | 34.5–2.1 |
| No. reflections | 30,758 (3,031) | 34,943 (4,282) | 39,092 (3,705) |
| $R_{work}/R_{free}$ | 0.21/0.26 | 0.22/0.25 | 0.18/0.22 |
| No. heavy atoms | 3,534 | 3,703 | 4,183 |
| Protein/DNA | 3,492 | 3,519 | 3,545 |
| $Sm^{3+}$, $K^+$ | 7 | 6 | 9 |
| Water | 36 | 173 | 625 |
| *B*-factors |  | 91 | 64 |
| Protein/DNA | 113.97 | 92 | 63 |
| $Sm^{3+}$, $K^+$ | 144 | 91.5 | 93.3 |
| Water | 110 | 77.3 | 69 |
| R.m.s. deviations |  |  |  |
| Bond lengths (Å) | 0.004 | 0.005 | 0.003 |
| Bond angles (°) | 0.56 | 0.60 | 0.54 |

Related to Figs 1 and 3.
One crystal was used for each mutant. Values in parentheses are for highest-resolution shell.

(Fig. 2 and Supplementary Fig. 3)[3,21,30]. The $+2$ and $+3$ nt of the 5′-flap were sandwiched between the main chain (residues 86–89 and residues 132–135) on one side of the channel and Leu97 on the other (Supplementary Fig. 2C) by non-sequence specific van der Waals contacts. The overall inverted flap orientation resembles the hFEN1-R100A structure with the $+1$ phosphate remaining within 7 Å of Arg104, but shifted towards Arg103, presumably due to Arg100 removal. Together, these substrate structures suggest that basic residues enable a phosphate steering mechanism, which we here define as electrostatic interactions that can dynamically position the phosphodiester backbone.

**Shifting of the scissile phosphate and the catalytic mechanism.**
In the hFEN1-D86N structure, we were surprised that the scissile phosphate was within catalytic distance of the active site while surrounding bases remained basepaired to the template strand (Fig. 3). This contradicts the prevailing hypothesis that surrounding bases must unpair for the scissile phosphate to move into the active site for incision[3]. Similarities and functionally-significant differences appeared on closer examination of hFEN1-D86N, hFEN1-R100A, and an earlier structure of FEN1-substrate with no 5′-flap or $+1$ phosphate (PDB code 3Q8L). In all three substrate structures, the dsDNA major groove is widened as it approaches the active site, and DNA bases flanking the scissile phosphate are stacked with one another, with the $+1$ base packed against Tyr40. However, the basepairing, the scissile phosphodiester bond location and the Tyr40 rotamer are distinctly different in the respective complexes, despite containing the same dsDNA sequence. In the 3Q8L structure, the DNA remained fully basepaired, and the scissile phosphodiester was positioned $\sim 6$ Å away from catalytic metals. In the hFEN1-R100A structure, the scissile phosphodiester bond was $\sim 4$–5 Å away from the metal ions, although $-1$ and $+1$ nts

have moved towards the active site and away from the template strand. The $-1$ and $-2$ nts display less base overlap (stacking), and the $+1$ and $-1$ nts are no longer hydrogen bonded to the template strand (4–6 Å apart).

In striking contrast, the scissile phosphodiester bond was directly coordinated to the one active site metal ion in the hFEN1-D86N structure (Fig. 3c). Furthermore, the $+1$ and $-1$ nts remained unexpectedly basepaired to the template strand, which is shifted relative to the other substrate structures via a dsDNA distortion surrounding the scissile phosphodiester (Fig. 3b and Supplementary Fig. 2D,E). There is no base stacking between $-1$ and $-2$ nts in the 5′-flap strand; instead, an unusual interstrand base stacking interaction occurs between the $-2$ nt of the 5′-flap strand and the template strand opposite of the $-1$ nt. We had hypothesized that unpairing of the $+1$ and $-1$ was required to move the scissile phosphate to within catalytic distance of the active site metals[3,21,31,32]. This new hFEN1-D86N substrate structure shows instead that basic residues can rotate dsDNA into the active site with basepairing intact (Fig. 3, Supplementary Movie 3). Moreover, since the DNA in 3Q8L, which was the furthest from the active site, lacked a 5′-flap or $+1$ phosphate, the DNA movements observed in the hFEN1-R100A and hFEN1-D86N structures are likely partly a consequence of either the 5′-flap and/or the $+1$ phosphate.

In concert with the DNA rotation in hFEN1-D86N, Tyr40 is in a different rotamer conformation from all other substrate or product bound or DNA-free hFEN1 structures (Fig. 3a,b and Supplementary Movie 3)[21,26]. This Tyr40 rotamer shift tracks duplex DNA rotation into the active site. The Tyr ring is fully stacked on the $+1$ base, and its side chain hydroxyl forms a hydrogen bond to the $+1$ phosphate. Notably, as duplex DNA is not shifted close to the catalytic metal in the R100A structure, this structure may represent a pre-reactive substrate form. Its Tyr40 stacks at a 50° angle with the $+1$ nt and resembles the other hFEN1 structures, suggesting that the Tyr40 rotamer is linked

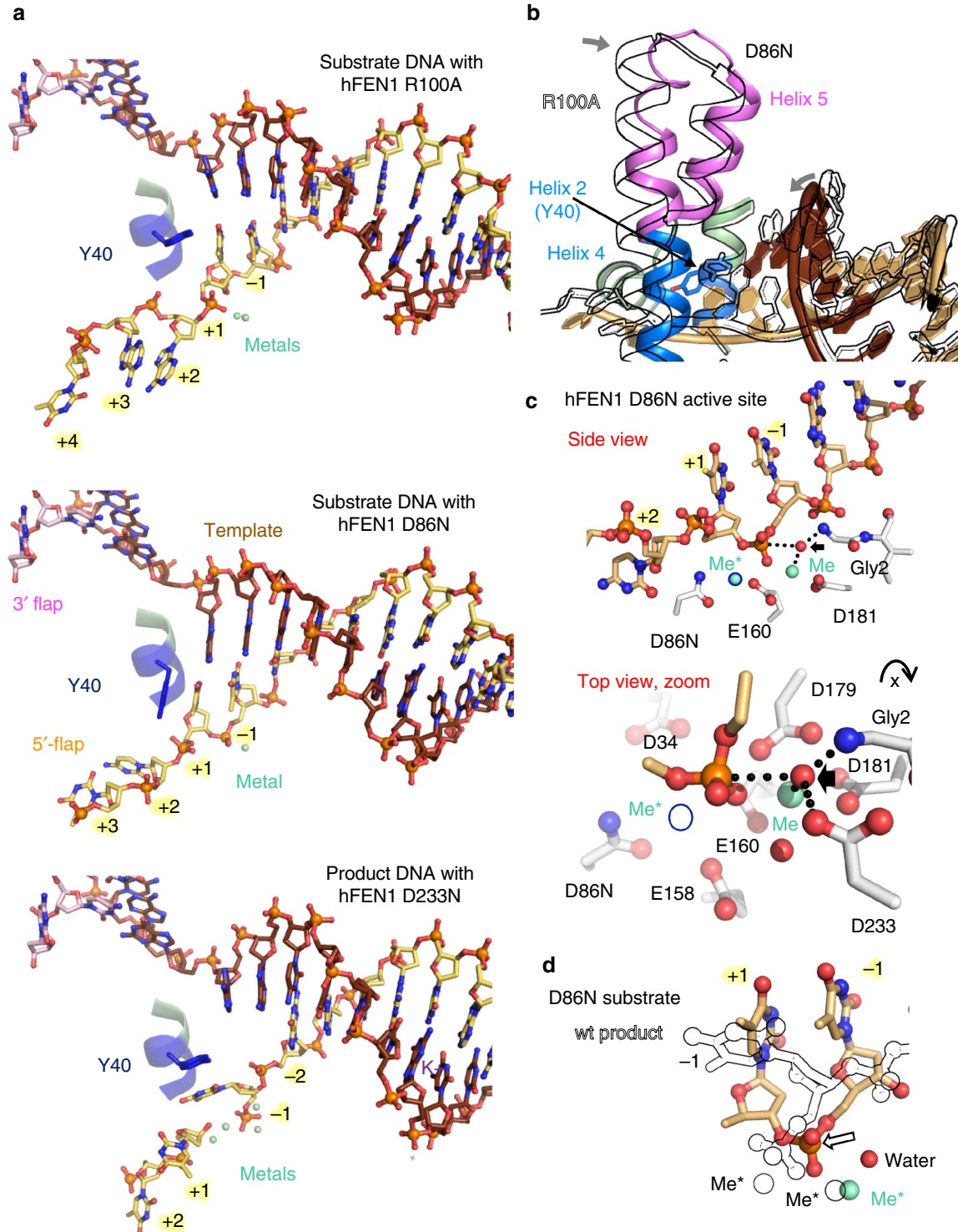

**Figure 3 | Three FEN1 crystal structures show threading through the capped gateway.** (**a**) DNA from hFEN1-D86N, hFEN1-R100A, and hFEN1-D233N structures showed threading not clamping. Tyr40 (stick model) changes its rotamer to track DNA movement through the gateway in the hFEN1-D86N structure. (**b**) A protein chain overlay between hFEN1-R100A (outline) and hFEN1-D86N (coloured) highlights how the helical gateway and cap and the DNA rotate closer together in the hFEN1-D86N complex. See Supplementary Movie 3. (**c**) The hFEN1-D86N active site revealed a water molecule positioned for linear attack on the scissile phosphate. Orthogonal views are shown. The 2nd metal position (outlined in black and denoted by Me*) is not observed in the hFEN1-D86N and is shown by overlaying the protein from the wt-product structure (PDB code 3Q8K). See also Supplementary Figs 1–4, and Supplementary Movie 4. (**d**) Protein chain overlay of hFEN1-D86N-substrate (coloured) and wt-product (outline, PDB code 3Q8K) structures shows that the scissile phosphate is shifted in the active site ~2 Å (demarked by arrow).

to shifting duplex DNA into a catalytic position. In the hFEN1-D86N-substrate structure, cap and gateway helices 4 and 5 are shifted 1–3 Å towards the dsDNA relative to all other

hFEN1 crystal structures with DNA (Fig. 3b and Supplementary Movie 3). The backbone of helix 2 (which contains Tyr40) does not change position.

Close examination of the hFEN1-D86N structure revealed a water molecule 3.3 Å from the scissile phosphate (Fig. 3c, Supplementary Fig. 4, and Supplementary Movie 4). This water is positioned for a linear attack on the scissile phosphate and for its evident activation by the catalytic metal and the Gly2 at the FEN1 N terminus, which was proposed to replace the 'third' metal in bacteriophage FEN[33]. Asp233 is 3 Å from the attacking water and contributes modestly to catalysis; the D233N mutant has 16-fold reduced but still substantial catalytic activity compared to mutants of other invariant carboxylates, such as D181A[21] and D86N (Supplementary Fig. 2B). When a second metal ion is modeled by overlay with the hFEN1-product structure (PDB code 3Q8K), the structure is reminiscent of the classical two-metal-ion catalysis[34]. Moreover, superfamily conserved and catalytically required[21] Lys93 and Arg100 sidechains point towards the scissile phosphodiester bond, poised to assist metal ion mediated hydrolysis. On the basis of the hFEN1-R100A structure, Arg100 is also likely essential for shifting of the scissile phosphate into direct contact with the catalytic metals. Notably, the scissile phosphate has moved ~1–2 Å between hFEN1-D86N-substrate and wt hFEN1-product (Fig. 3d). An analogous metal movement into more optimal coordination geometry in an RNaseH-product structure was proposed to favour product formation[35]. We cannot exclude a possible third metal ion as time-resolved experiments on other enzymes show metals ions can appear and disappear during reaction[36–40].

Together these structures reinforce and extend biochemical data that suggest that FEN1 checks for the ss 5′-flap by threading it through a tunnel formed between the active site and capped gateway helices (Fig. 1d)[24,41]. The substrate structures imply the 5′-flap is (1) electrostatically steered through the capped gateway by conserved basic residues in the gateway and cap and (2) positioned in an inverted orientation.

**Biochemically testing phosphate steering.** If the gateway/cap region basic residues steer the phosphodiester backbone as implied by the structures, then their mutation should affect 5′-flap substrate incision rates. On the basis of the hFEN1-R100A structure, we mutated three basic residues (Arg103, Arg104 and Lys132) positioned to guide the phosphodiester backbone and stabilize the inverted ssDNA orientation (Fig. 1d). We also mutated Arg129, which is adjacent the other residues and could act in steering. When the helical cap is structured, Arg129 makes a long-range electrostatic interaction with a phosphate of the template strand[21], a distance shortened in hFEN1-D86N by template strand relocation. Strikingly, these four basic residues are conserved across all FEN1s including yeast and archaeal, except for the less-specific phage 5′ nucleases (Supplementary Fig. 3). As the helical gateway and cap regions are flexible before productive DNA binding[22,24,26], specific interactions would seem unlikely during the flap threading process but electrostatic guidance is possible. Notably, as these side chains range from 10 to 19 Å from the target phosphate, they are unlikely to impact FEN1 activity by aspects other than electrostatic guidance and substrate-positioning. To test this idea, we mutated them to alanine or glutamate to either remove the attractive positive charge or provide a repulsive charge, respectively.

These charge mutations all reduced specific incision activity on a 5′-flap substrate, S5,1 (Fig. 4a), indicating an important functional role. Under multiple turnover conditions, single mutations R103A and K132A moderately decreased the reaction rate relative to wt hFEN1 by 3- and 5-fold, respectively, whereas a 20-fold decrease was observed with either R104A or R129A (Fig. 4b and Supplementary Fig. 5A,B). These rate decreases are consistent with a single residue electrostatic guidance interaction[42]. Double mutant R104AK132A showed an additive effect with 200-fold reduced activity and, significantly, the corresponding repulsive mutant R104EK132E was far more severely compromised with a rate reduction of 11,000 compared to the wt enzyme. Importantly, the substrate dissociation constant ($K_d$) for each of these double mutants was only modestly raised (Supplementary Fig. 6). This suggests deficient substrate positioning, not poor binding, as the major contributing factor to diminished activity. Similarly, double mutants R103AR129A or R103ER129E showed reductions in reactivity of 70- or 5,000-fold, respectively,

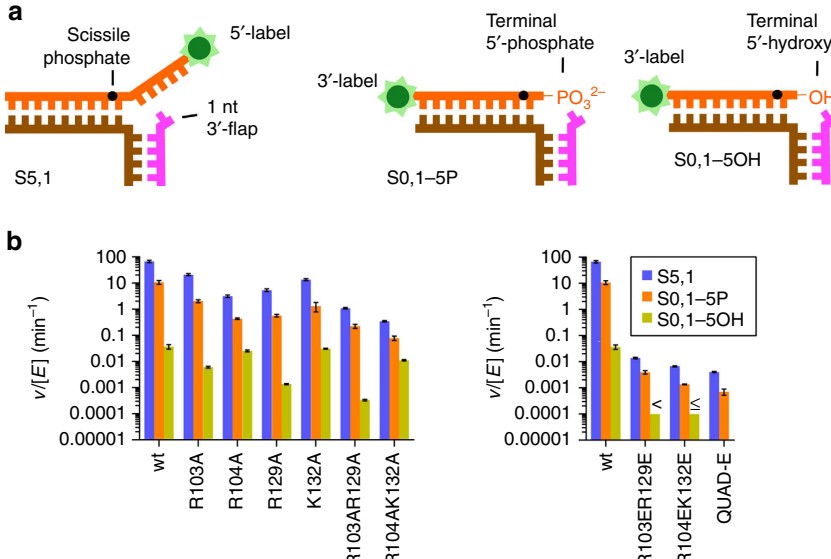

**Figure 4 | Phosphate steering residue mutants show reduced activity.** (**a**) Schematic of substrates used with fluorescent positions. (**b**) Comparison of multiple turnover rates for cleavage of each substrate at 50 nM. Reaction rates for glutamate mutants with S0,1-5OH were too slow to measure accurately, so threshold values are indicated. See also Supplementary Figs 1, 5 and 6. Error bars are shown as a function of s.e.m., with replicate number given in Supplementary Fig. 5E.

without any substantial effect on $K_d$. Analogous trends in rate effects were observed under single turnover kinetic conditions (Supplementary Fig. 5C,D).

Mutating all four gateway/cap residues to glutamate ('QUAD-E' mutant) severely impaired activity (18,000-fold slower than wt FEN1). Strikingly, the $K_d$ increased only 17-fold showing the enzyme was folded and capable of substrate binding. This large rate decrease is remarkable for mutation of residues not acting in catalysis and distant from the active site: it resembles the penalty for streptavidin added to 5'-biotinylated substrates, which would prevent 5'-flap gateway/cap threading[24].

If the FEN1 basic cap residues are primarily required for ss 5'-flap steering, then their mutation should not be deleterious to incision activity on an exonucleolytic substrate lacking a 5'-flap but with a 5'-phosphate (S0,1-5P; Fig. 4a). This substrate was hydrolysed sevenfold more slowly than the DF S5,1 by wt hFEN1 (Fig. 4b and Supplementary Fig. 5A,B) showing that threading the 5'-flap facilitates access to the catalytically competent conformation, as well as being a key mechanism in substrate selection. For reaction rates expressed relative to wt hFEN1 to normalize for this sevenfold difference, the gateway mutants all proved similarly defective on both the exonucleolytic (S0,1-5P) and endonucleolytic (S5,1) substrates (relative rates given in Supplementary Fig. 4B). These results unmask a key universal role for +1 phosphate steering in the FEN1 incisions of both exonucleolytic and endonucleolytic substrates (since this phosphate is present in both substrates).

Given the results with the exonucleolytic substrate and the observation that DNA movement towards the active site required a +1 phosphate[43], we reasoned that some basic residues were electrostatically interacting with the +1 phosphodiester of dsDNA to facilitate this movement[44]. To test this hypothesis, we measured reaction rates with an analogous exonucleolytic substrate lacking the 5'-phosphate at the +1 position (S0,1-5OH; Fig. 4a). This substrate was bound 20-fold more weakly and incised 300-fold more slowly than S0,1-5P by wt hFEN1 (Fig. 4b, Supplementary Figs 4 and 5). These data indicate that 5'-phosphate (+1 phosphate) interactions stabilize the enzyme-substrate complex and contribute to catalysis. Combined and individual mutations of R103A and R129A all decreased incision rates of S0,1-5OH analogously to the other substrates. However, R104A, K132A and R104AK132A all processed S0,1-5OH at a similar rate to wt hFEN1. These results imply that the +1 phosphate group functionally interacts with Arg104 and Lys132, consistent with the phosphate steering hypothesis, but that Arg103 and Arg129 (along with Arg100 and Lys93) have long-range interactions to other parts of the DNA substrate, including the scissile phosphate itself.

**A role for phosphate steering in genome stability.** To test the biological importance of the basic cap and gateway residues, we made equivalent mutations in the helical gateway/cap region of Rad27, the *S. cerevisiae* homolog of hFEN1, to analyse their role in genome integrity *in vivo*. Alanine or glutamate mutations were introduced at Rad27 Arg104, Arg105, Arg127 and Lys130; equivalent to hFEN1 Arg103, Arg104, Arg129 and Lys132, respectively (Fig. 5a). Growth characteristics of double- or quadruple-mutant yeast strains were compared to wild type *RAD27* (wt) strain and *rad27-D179A* (corresponding to human D181A whose incision rate is given in Supplementary Fig. 2B), a severely catalytically impaired mutant, which displays an equivalent phenotype to the *rad27* null strain (that is, sensitivity to hydroxyurea, a replication inhibitor and DNA-damaging UV light[1,45]).

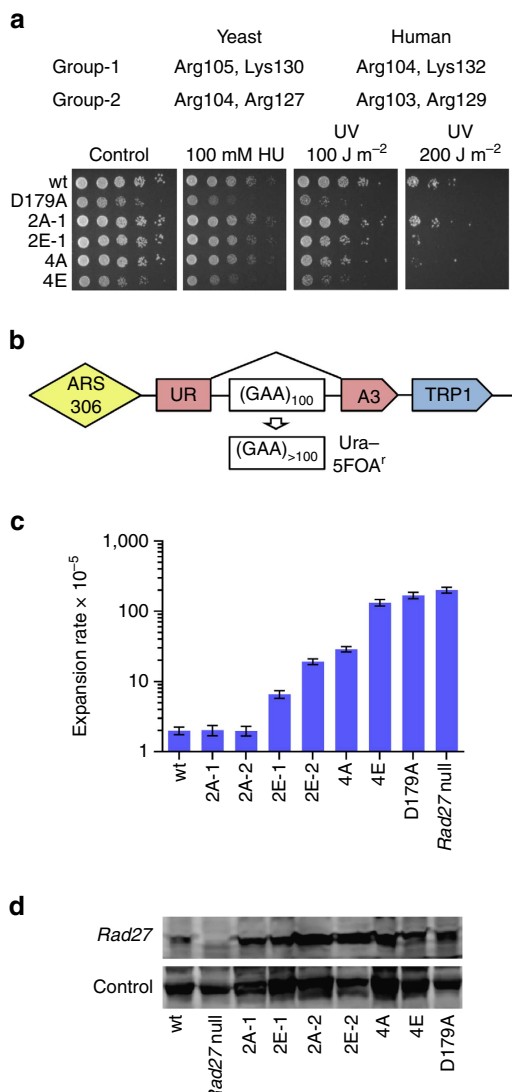

Figure 5 | **Phenotypic and DNA repeat expansion defects of *Rad27* basic cap residue mutations in yeast *S. cerevisiae*.** (**a**) Table of the tested basic residues in yeast (and their human counterpart) and spot-test (serial fivefold dilutions) for yeast growth with and without exposure to hydroxyurea (replication inhibitor) or UV light (DNA-damaging). (**b**) Experimental system to measure the rates of large-scale repeat expansions in yeast. The $(GAA)_{100}/(TTC)_{100}$ repeat is incorporated into the intron of an artificially split *URA3* gene. Addition of ≥10 extra repeats inhibits reporter's splicing, which allows cells with repeat expansions to grow on 5-FOA-containing media. (**c**) Effect of active site control and phosphate steering mutations in the *RAD27* gene on repeat expansion rates (error bars represent 95% confidence intervals of calculated expansion rates). (**d**) Rad27 protein expression was not substantially altered in the mutated strains. See also Supplementary Fig. 7 and Supplementary Table 1.

Even without exogenous treatment, quadruple glutamate mutant (Fig. 5a) showed growth inhibition resembling that for the active site mutant *rad27-D179A*. Replication stress induced by hydroxyurea greatly accentuated this effect. Moderate UV irradiation ($100\,J\,m^{-2}$) was strongly deleterious to both strains, and higher-dose irradiation ($200\,J\,m^{-2}$) further revealed UV sensitivity for the double glutamate (2E-1; R105EK130E) and quadruple alanine (4A) mutants (Fig. 5a). Thus, electrostatic interactions of gateway/cap basic residues with DNA are critical for flap endonuclease biological function, with particular

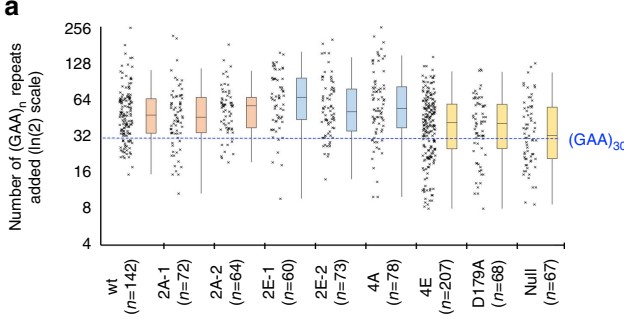

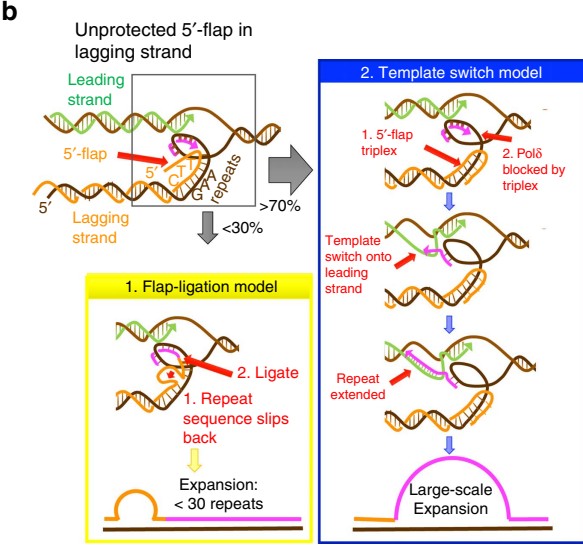

**Figure 6 | FEN1 phosphate steering is essential for lagging strand precision at DNA repeats. (a)** Graphed distributions of repeat expansion lengths shows that the majority of expansions in the wt, phosphate steering and D179A Rad27 mutants are >30 repeats. The numbers of added repeats in each strain are shown as scatter plots alongside box-and-whisker plots with 5 and 95% whiskers. The number of colonies tested are given in the parentheses. **(b)** Two models for repeat expansions driven by the presence of an unprocessed 5'-flap. In model 1 (left panel) the repeat on the 5'-flap ligates to the 3'-end of the oncoming Okazaki fragment followed by its equilibration into a loop. After the next round of replication, up to ∼30 repeats can be added (see text for details). In model 2 (right panel), the 5'-flap folds back forming a triplex, which blocks Pol(δ) DNA synthesis along the lagging strand template and promotes its switch to the nascent leading strand. This template switch mechanism explains the accumulation of large-scale repeat expansions >30 repeats.

deleterious effects on cells under replication stress and/or with damaged DNA.

Rad27 inactivation in yeast stimulates expansion of trinucleotide repeats relevant to human disease[46–48]. We therefore tested the effect of phosphate steering mutations on expansion rates of $(GAA)_n$ repeats using our system (Fig. 5b), which contains a $(GAA)_{100}$ tract situated in the intron of a *Ura3* reporter gene[49,50]. Addition of 10 or more repeats to the $(GAA)_{100}$ tract effectively blocks splicing, resulting in gene inactivation and rendering the yeast resistant to 5-fluoroorotic acid (5-FOA). The repeat expansion rates in the *rad27* knockout and in the severely catalytically impaired D179A active site metal ligand mutant was increased by ∼100-fold compared to wt (Fig. 5c,d). Strikingly for a non-active site mutant, phosphate steering 4E mutant exhibited a quantitatively similar phenotype. The double glutamate (2E-1, 2E-2) and 4A mutants showed intermediate (∼10-fold) increases

in repeat expansion rates. These results match growth characteristics of these mutants and emphasize the role of electrostatic interactions of the gateway basic residues with DNA in repeat-mediated genome instability.

Ligation of unprocessed 5'-flaps to the 3'-end of the approaching Okazaki fragment is proposed to cause the elevated repeat expansions in Rad27 mutants[48,51,52]. In this scenario, one expects added repeat lengths to be relatively short: less than the size of an Okazaki fragment. In fact, the major mutations caused by disruption of the *RAD27* gene in yeast were repeat-related expansions of 5–108 bases[53]. Recently, the median size of the unprocessed 5'-flap in *S. pombe FEN1* knockout was measured as 89 nts[54]. Given these numbers, the median expansion size of GAA repeats in our experimental system should be ∼30 repeats in Rad27 mutants.

To define the size distribution of expansion products, we measured the scale of repeat expansions in wt and Rad27 mutants described above via PCR (Fig. 6a). In the wt strain, median expansion size corresponded to 47 triplets[49]. The *rad27* knockout was different: median expansion size was 32 repeats, and Kolmogorov–Smirnov (KS) comparison confirms a significant difference from the wt strain ($P < 0.001$), which agrees with known flap size in *FEN1* knockouts[54]. The expansion scale in near-catalytic-dead (D179A) and 4E Rad27 mutants lies between the wt and knockout mutant: the median is 40 repeats and KS shows significant difference from wt ($P < 0.05$). Finally, the scale of expansions in 2E and 4A mutants is greater than wt with medians from 50 to 66 added repeats. Thus, the 100-fold increase in expansions (Fig. 5c) in phosphate steering mutants cannot be explained by an increase in small-scale expansions alone (caused by simple 5'-flap ligation), but is a consequence of larger expansions. Thus, most expansions in the Rad27 phosphate steering mutants originate via mechanisms distinct from simple 5'-flap ligation (see Discussion). Overall, these Rad27 results suggest that functional phosphate steering of 5'-flaps and dsDNA is vital for genome integrity: in promoting normal growth, in response to DNA damage, and in preventing trinucleotide repeat expansions.

## Discussion

We sought to understand the mechanism whereby FEN1s binds and precisely incises ss-dsDNA junctions yet excludes hydrolysis of continuous DNA substrates, reasoning that this specificity was key to FEN1 functions during replication and repair. These investigations resolve controversies and improve our understanding of how FEN1-DNA interactions provide specificity and genome stability.

First, elucidation of a 5'-flap DNA threaded through the helical gateway/cap answers a longstanding question in eukaryotic FEN1 function and explains the selection of 5'-flap substrates with free 5'-termini. Although threading occurs in other enzymes, phosphate steering and inverted threading are extraordinary. For example, bacteriophage T5 5'-nuclease threads substrates[29], but positions the 5' flap primarily through hydrophobic interactions to the 5' flap nucleobases. The phosphodiester is closer to the metals than the nucleobases, consistent with its lower incision site specificity and tendency to cleave within the ssDNA 5'-flap. In other enzymes, threading selects for free ss 5'-termini that will undergo incision and there is no inversion. However, FEN1 preserves, rather than degrades, the threaded nucleic acid.

Second, our results uncover an essential function in FEN1 specificity and catalysis for phosphate steering, which we define as electrostatic interactions that dynamically control the phosphodiester backbone. The parallel effects of steering mutations on either endonucleolytic or exonucleolytic reactions

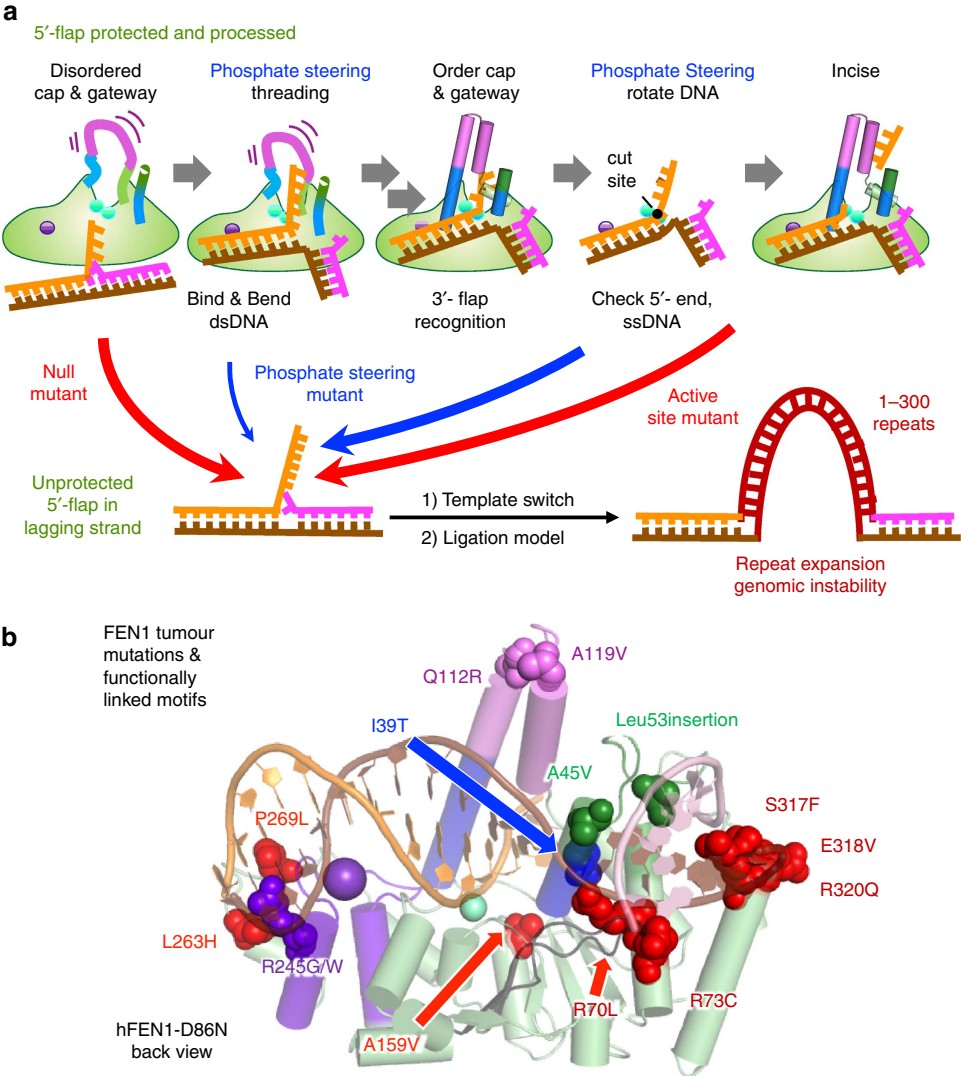

**Figure 7 | Multiple motifs for FEN1 substrate recognition and hydrolysis ensure accurate incision activity and prevent genomic instability.**
(**a**) Schematic model of the FEN1 mechanism emphasizing the functional role of phosphate steering in the dynamic processes of 5′-flap inverted threading and shifting of the duplex DNA towards the catalytic metals. (**b**) Tumour-associated mutations from breast, lung, skin, kidney, colorectal, ovarian and testicular cancers map to functionally-important structural motifs: dsDNA binding (P269L, L263H, R245G/W, R70L and R73G), 3′-flap binding (Leu53ins, A45V, S317F, E318V and R320Q), helical gateway/cap (I39T, Q112R and A119V) or active site (A159V).

(that is, on substrates with or without a 5′-flap) indicated involvement of basic gateway/cap residues in a rate-limiting step in the FEN1 catalytic pathway, that is, in moving the target phosphodiester bond from the ss-ds junction onto catalytic metal ions. Thus, phosphate steering may act in orienting the ss 5′-flap during threading (negative design to avoid off-target reactions) and moving the scissile phosphate into catalytic distance of the metals (positive design to enhance target reactions) (Fig. 7a). Notably, steering residue Arg104, is semiconserved throughout the superfamily suggesting that phosphate positioning occurs in other members.

Third, the proposed requirement for double base unpairing for the dsDNA to reach the active site metal ions[3] needs re-evaluation. Our observation of basepaired DNA contacting an active site metal ion with a water molecule positioned for in-line attack, would generate the arrangement for 'two-metal-ion' catalysis. This basepaired catalytically competent conformation appears at odds with spectroscopic characterization of FEN1 and GEN1 substrate complexes[19,32,44], and the inability of FEN1 to

process duplexes cross-linked at the terminal basepair[31,55], consistent with an unpairing mechanism. Yet, the DNA distortion seen in structures here (Supplementary Fig. 2E) provides an alternative explanation implying dsDNA can remain basepaired and roll onto the active site metal ions aided by Tyr40 rotation and by positive side chains on the helical gateway and cap.

Whereas replication fidelity is canonically based on sequence, it furthermore depends on sequence-independent specificity in FEN1. Importantly, structural elements critically involved in FEN1 function, including phosphate steering and inverted threading, require key residues distant from the active site metal ions. Indeed, clinically relevant FEN1 mutations compiled by The Cancer Genome Atlas (TCGA) and others[5,56,57] map to these structural elements (Fig. 7b). So, although tumour mutation data has been called 'a bewildering hodgepodge of genetic oddities'[58], for FEN1, there is a clear link of structurally-mapped mutations to compromised function, genomic instability and cancer. Although these mutations may retain nuclease activity, even

tiny off target activity risks toxicity and genomic instability, and replication mutations account for two-thirds of the mutations in human cancers[59].

We uncovered a role for phosphate steering in triplet (GAA)$_n$ repeat expansions, that also implicates template switching from the lagging strand due to FEN1 defects. Most expansions in Rad27 phosphate steering mutants were large-scale (>30 repeats; Fig. 6a) which is difficult to explain by the canonical flap-ligation model for repeat instability[46,48,51,60]. In this model, an unprocessed 5′-flap is ligated to the 3′-end of the approaching Okazaki fragment (Fig. 6b, left), limiting the length of expansions to the size of those flaps. Recently, the median size of the 5′-flap in a FEN1 knockout was found to be 89 nts[54], that is, ∼30 triplet repeats. Since median expansion size in phosphate steering mutants is >30 repeats, we propose that besides a flap-ligation model, a template switch between nascent repetitive strands occurs as a replication fork stumbles through the repeat sequence[50] (Fig. 6b, right). Unprocessed (TTC)$_n$ 5′-flaps of the Okazaki fragments may form a stable triplex[61] with the downstream repetitive run. This could block displacement synthesis by the lagging strand polymerase[62] and prompt it to search for a new template. Large-scale repeat expansions would then occur when the polymerase switches template—continuing DNA synthesis along the nascent leading strand. As a starting repeat gets longer, larger expansions become feasible, consistent with the progressive increase in expansion amplitudes with the length of original repeat tract, as observed in human pedigrees[63].

The profound stimulation of large-scale expansions in the phosphate steering mutants unexpectedly sheds light on the molecular mechanism of template switching. A priori, either a nascent leading strand can switch onto the nascent lagging strand to use it as a template[49,50], or the nascent lagging strand can switch onto the nascent leading strand serving as a template[47]. Since it is the lagging strand synthesis and specifically Okazaki fragment maturation that are unraveled in Rad27 mutants, the sheer magnitude of their effects on large-scale repeat expansions implies that the lagging strand likely switches onto the nascent leading strand accounting for the repeat instability, and this merits further biochemical investigation. Another question emerges from biochemical studies of FEN1 functions during long-patch base excision repair where expansions occur on dysregulation of DNA handoffs from polymerase β to FEN1 (ref. 64), which suggests studies to investigate whether phosphate steering may prevent expansions during long-patch repair.

In summary, we find FEN1 phosphate steering energetically promotes dsDNA rotation into the active site and inverted threading of the 5′-flap to enforce efficiency and fidelity in replication and repair. Interestingly, elevated FEN1 expression safeguards against repeat instability in somatic tissues[6]. Phosphate steering mutations could thus be the trans-modifiers of repeat expansions during either somatic, or intergenerational transmissions in human disease[65]. Moreover, as the basic residues implicated in phosphate steering are largely conserved in the 5′-nuclease superfamily, control over the +1 and −1 phosphates may be a superfamily-conserved mechanism.

## Methods

**Site-directed mutagenesis.** Plasmids for expression of mutant proteins were prepared from either the pET29b-hFEN1Δ336(wt) or pET28b-hFEN1-(His)$_6$ constructs, as indicated above, following the protocol outlined in the QuikChange site-directed mutagenesis kit (Agilent Technologies, Inc.). Mutagenic primers were purchased from Fisher Scientific, with desalting, then reconstituted in ultrapure water and used as supplied. Mutagenic primer sequences were as follows: D86N, 5′-ggcggcttgccattaaagacatacacgggct-3′ and 5′-aagcccgtgtatgtctttaatggcaagccgcc-3′; R103A, 5′-caaacgcagtgaggcgcgggctgaggca-3′ and 5′-tgcctcagcccgcgcctcactgcgtttg-3′; R103E, 5′-ccaaacgcagtgaggagcgggctgaggcag-3′ and 5′-ctgcctcagcccgctcctcactgcgtttgg-3′; R104A, 5′-ctctgcctcagccgccccgctcactgcgt-3′ and 5′-acgcagtgagcgggcggctgaggc

agag-3′; R104E, 5′-acgcagtgagcgggaggctgaggcagag-3′ and 5′-ctctgcctcagcctcccgctcactgcgt-3′; R129A, 5′-ttagtgaccttcaccagcgccttagtgaattttttccacctc-3′ and 5′-gaggtggaaaaattcactaaggcgctggtgaaggtcactaa-3′; R129E, 5′-ttagtgaccttcaccagctccttagtgaattttttccacctc-3′ and 5′-gaggtggaaaaattcactaaggagctggtgaaggtcactaa-3′; K132A, 5′-cactaagcggctggtggcggtcactaagcagcac-3′ and 5′-gtgctgcttagtgaccgccaccagccgcttagtg-3′; K132E, 5′-gctgcttagtgacctccaccagccgcttagt-3′ and 5′-actaagcggctggtggaggtcactaagcagc-3′; R103ER104E, 5′-gcttctctgcctcagcctcctcctcactgcgtttggcca-3′ and 5′-tggccaaacgcagtgaggaggaggctgaggcagagaagc-3′; R129EK132E, 5′-gtgctgcttagtgacctccaccagctccttagtgaattttttccacc-3′ and 5′-ggtggaaaaattcactaaggagctggtggaggtcactaagcagcac-3′.

**Protein expression.** Plasmids encoding R100AΔ336 and D233NΔ336 human FEN1 for crystallography were generated by site-directed mutagenesis from the pET29b-hFEN1Δ336(wt) construct bearing a PreScission protease site and (His)$_6$-tag after residue 336 of the wt sequence[21]. Full length wt hFEN1 was encoded using the pET28b-hFEN1-(His)$_6$ vector reported previously[12], and all reported mutants were generated from this by site-directed mutagenesis. Proteins were expressed in Rosetta (DE3)pLysS competent cells grown in 2 × YT media or Terrific Broth to an OD$_{600}$ of 0.6–0.8 at 37 °C then induced by addition of 1 mM IPTG, followed by incubation at 18 °C for 18–24 h. Cells were collected by centrifugation at 6,000 g/4 °C, washed with PBS, then resuspended in buffer IMAC-A1 (20 mM Tris pH 7.0, 1.0 M NaCl, 5 mM imidazole, 0.02% NaN$_3$, 5 mM β-mercaptoethanol supplemented with SIGMAFAST protease inhibitor tablets and 1 mg ml$^{-1}$ chicken egg white lysozyme). Each suspension was kept on ice for 2 h then stored frozen at −20 °C until further processing, as detailed below.

**Purification of hFEN1 D86NΔ336 and R100AΔ336 and D233NΔ336.** All steps were carried out at 4 °C. Chromatography was on an ÄKTA system with flow rate of 5.0 ml min$^{-1}$ unless stated otherwise. Columns were from GE Healthcare, unless stated otherwise. Frozen lysates were thawed on ice and homogenized by sonication. Next, 0.1 volume of a 10% v/v TWEEN 20 solution was added. The mixture was clarified by centrifugation at 30,000 g for 30 min. Supernatant was loaded onto a Ni-IDA affinity column, which was then washed with 5 column volumes (CV) of buffer IMAC-A1, 5 CV of buffer IMAC-A2 (20 mM Tris pH 7.0, 0.5 M NaCl, 40 mM imidazole, 0.02% NaN$_3$, 0.1% v/v TWEEN 20, 5 mM β-mercaptoethanol). FEN1 was eluted with 5 CV of buffer IMAC-B1 (250 mM imidazole pH 7.2, 0.5 M NaCl, 0.02% NaN$_3$, 5 mM β-mercaptoethanol). Pooled fractions were diluted 1:5 with water and then loaded onto a HiPrep Heparin FF 16/10 column. The column was washed with 5 CV buffer HEP-A1 (25 mM Tris pH 7.5, 1 mM CaCl$_2$, 0.02% NaN$_3$, 20 mM β-mercaptoethanol). FEN1 was eluted with a linear gradient of 100% HEP-A1 to 100% HEP-A2 (25 mM Tris pH 7.5, 1 mM CaCl$_2$, 1.0 M NaCl, 0.02% NaN$_3$, 20 mM β-mercaptoethanol) in 20 CV. Pooled FEN1 fractions were diluted by slow addition of two volumes of 3.0 M (NH$_4$)$_2$SO$_4$ at 4 °C. The solution was loaded onto a HiPrep Phenyl FF (high sub) 16/10 phenylsepharose column. The column was washed with 7 CV buffer P/S-B1 (25 mM Tris pH 7.5, 2.0 M (NH$_4$)$_2$SO$_4$, 2 mM CaCl$_2$, 0.02% NaN$_3$, 20 mM β-mercaptoethanol). FEN1 was eluted with a gradient of 100% P/S-B1 to 100% P/S-A1 (25 mM Tris pH 7.5, 10% v/v glycerol, 1 mM CaCl$_2$, 0.02% NaN$_3$, 20 mM β-mercaptoethanol) in 20 CV. Pooled fractions were concentrated to ∼7 ml using an Amicon stirred cell (Merck Millipore), then passed through 5 × 5 ml HiTrap Desalting columns arranged in tandem, injected in 1.5 ml portions. The desalting columns were equilibrated in 1 × TBS supplemented with 1 mM EDTA and 1 mM DTT, and eluted with the same buffer. Combined protein-containing eluent (35–40 ml) was treated with PreScission protease (20 μl of activity 10 U μl$^{-1}$) and incubated at 4 °C overnight. Complete cleavage of the (His)$_6$ tag was verified by SDS-PAGE, then the protein solution concentrated to 5 ml using a Vivaspin 20 Centrifugal Concentrator (10,000 MWCO). A final purification step at a 0.5 ml min$^{-1}$ flow rate with a Sephacryl S-100 HR column, equilibrated with 2 CV of 2 × SB (100 mM HEPES pH 7.5, 200 mM KCl, 2 mM CaCl$_2$, 10 mM DTT, 0.04% NaN$_3$). FEN1 fractions were pooled and the protein concentration determined by A$_{280}$, using the calculated OD$_{280}$. The solution was concentrated to >200 μM using a Vivaspin 20 Centrifugal Concentrator (10,000 MWCO). Finally, the solution was mixed 1:1 v/v with cold glycerol, placed on a roller mixer until homogenous, then divided into 1 ml aliquots and stored as a 100 μM stock solution at −20 °C.

**Crystallography of mutant FEN1-DNA complexes.** hFEN1 mutants were crystallized with DF substrates (S5,1) or (S4,1) of slightly different sequence (desalted purity from IDT, Supplementary Fig. 1). hFEN1-D86NΔ336 (19 mg ml$^{-1}$) was mixed in volumetric ratio 1:2:1 with 4.25 mM SmSO$_4$, and 1.3 mM substrate S5,1-D86N. This mixture was in turn combined 1:1 with 12% mPEG 2,000, 20% saturated KCl, 5% ethylene glycol, 100 mM HEPES pH 7.5. Crystals were collected after 5 days at 15 °C. hFEN1-R100AΔ336 (19 mg ml$^{-1}$) was mixed in volumetric ratio 1:2:1 with 3.75 mM SmSO$_4$, and 1.3 mM substrate S4,1-R100A. This mixture was in turn combined 1:1 with 22% mPEG 2000, 20% saturated KCl, 5% ethylene glycol, 100 mM HEPES pH 7.5. Crystals were collected after ∼3 weeks at 15 °C. hFEN1-D233NΔ336 (8.2 mg ml$^{-1}$) with 1.6 mM SmSO$_4$, 0.25 mM substrate S4,1-D233N was mixed 1:1 with 24% mPEG 2000, 20% saturated KCl, 5% ethylene glycol, 100 mM HEPES pH 7.5. hFEN1-D86N data was collected at 0.98 Å (SSRL beamline 12-2) and processed with HKL2000. hFEN1-R100A data was collected at 0.98 Å (SSRL beamline 9-2) and processed with XDS.

hFEN1-D233N data was collected at 1.12 Å (ALS beamline 12.3.1) and processed with HKL2000. hFEN1-D86N, hFEN1-R100 and hFEN1-D233N crystals diffracted to 2.8, 2.65 and 2.1 Å, respectively. Structures were solved by molecular replacement using PHASER[66] with human FEN1 protein as the search model and refined in PHENIX[67] with rounds of manual rebuilding in COOT[68]. For hFEN1-R100A, we refined the model using higher diffraction data to 2.1 Å, based on the theory that cutting off resolution at an arbitrary point leads to series termination errors. Flexible regions became more visible and we could follow the path of the 5′-flap more easily. The R and $R_{\text{free}}$ measures dropped substantially. We used a higher resolution structure (PDB code: 3Q8K) for reference in refinement. For the three structures, anomalous differences from the $Sm^{3+}$ atoms were used in refinement and modelling. In the active sites of the hFEN1-D86N, hFEN1-R100A and hFEN1-D233N structures there were, respectively, one, three and four $Sm^{3+}$ atoms, with partial occupancy. For all structures, there were no Ramachandran outliers. For hFEN1-D86n, 95% were favoured and 5% were allowed. For hFEN1-R100A, 96% were favoured and 4% were allowed. For hFEN1-D233N, 98% were favoured and 2% were allowed. Structure figures were created in PyMol (Schrödinger, LLC). Movies were created in Chimera[69].

**Protein purification of full-length FEN1 proteins.** All steps were carried out using an ÄKTA FPLC system at 4 °C, at a flow rate of 5.0 ml min$^{-1}$ unless stated otherwise. Frozen/thawed lysates were loaded onto a Ni-IDA column, followed by washing with 4 CV buffer IMAC-A1, 4 CV buffer IMAC-A2, a gradient of 100% IMAC-A2 to 100% IMAC-B1 in 2 CV, then 4 CV IMAC-B1. Pooled fractions were diluted 1:1 with 20 mM β-mercaptoethanol and loaded onto a 5 ml HiTrap Q FF column to remove nucleic acid contamination, with a 20 CV elution gradient from 0 to 1.0 M NaCl in 20 mM Tris pH 8.0, 1 mM EDTA, 0.02% NaN$_3$, 20 mM β-mercaptoethanol. The flow-through containing FEN1 was diluted 1:4 with 20 mM β-mercaptoethanol and passed through the HiPrep Heparin FF 16/10 column as above. The purified FEN1 was exchanged into 2 × SB using a HiPrep 26/10 Desalting column, concentrated and prepared for storage as detailed above. Proteins requiring further purification (wt hFEN1 and D233N) were passed through the HiPrep Phenyl FF (high sub) 16/10 phenylsepharose column, as above. Protein-containing fractions were pooled and concentrated to 5 ml using an Amicon stirred cell, subjected to gel filtration and prepared for storage as outlined above.

**Oligonucleotide synthesis.** The DNA oligonucleotides used for crystallization (Supplementary Fig. 1) were purchased from IDT as desalted oligonucleotides. They were resuspended in 10 mM HEPES 7.5, 50 mM KCl, 0.5 mM EDTA and annealed at ∼1–2 mM. The DNA oligonucleotides used to construct the kinetic substrates (Supplementary Fig. 1) were purchased from DNA Technology A/S (Denmark) with HPLC purification. Except for E1 and E2 (Supplementary Fig. 1A), the oligonucleotides as supplied were reconstituted in ultrapure water and concentrations of stock solutions determined using calculated extinction coefficients (OD$_{260}$). Oligonucleotides E1 and E2 required additional HPLC purification, which was carried out using an OligoSep GC cartridge (Transgenomic; #NUC-99–3860) using buffers A (100 mM triethylammonium acetate pH 7.0, 0.025% v/v acetonitrile) and B (100 mM triethylammonium acetate pH 7.0, 25% acetonitrile) and a gradient of 5–50% B over 18 min, at 50 °C and a flow rate of 1.5 ml min$^{-1}$. Purified oligonucleotide in solution was loaded onto a 5 ml HiTrap DEAE FF column equilibrated with 3 CV of buffer C (10 mM Tris pH 7.5, 100 mM NaCl, 1 mM EDTA, 0.02% NaN$_3$). The column was washed with a further 3 CV of buffer C, then eluted using a step gradient of 100% buffer C-100% buffer D (10 mM Tris pH 7.5, 1.0 M NaCl, 1 mM EDTA, 0.02% NaN$_3$) in 3 CV. Fractions containing DNA were desalted into ultrapure water using NAP-25 columns. Desalted samples were dried then reconstituted as above. DNA constructs were annealed in 1 × FB (50 mM HEPES pH 7.5, 100 mM KCl) for at least 5 min at 95 °C, then left at ambient temperature for 30 min.

**FRET binding assay.** Values for $K_d$ were obtained using sequential titration of the appropriate enzyme into a 10 nM solution of the appropriate DNA construct, according to the reported protocol[44]. FRET efficiencies (E) were determined using the (ratio)$_A$ method by measuring the enhanced acceptor fluorescence at 37 °C. The steady state fluorescent spectra of 10 nM non-labelled (NL) trimolecular, donor-only labelled (DOL) and doubly labelled (DAL) DNA substrates (Supplementary Fig. 1A,B) were recorded using a Horiba Jobin Yvon FluoroMax-3 fluorometer. For direct excitation of the donor (fluorescein, DOL) or acceptor (TAMRA, AOL), the sample was excited at 490 nm or 560 nm (2 nm slit width) and the emission signal collected from 515–650 nm or 575–650 nm (5 nm slit width). Emission spectra were corrected for buffer and enzyme background signal by subtracting the signal from the NL DNA sample. In addition to 10 nM of the appropriate DNA construct, samples contained 10 mM CaCl$_2$ or 2 mM EDTA, 110 mM KCl, 55 mM HEPES pH 7.5, 0.1 mg ml$^{-1}$ bovine serum albumin and 1 mM DTT. The first measurement was taken before the addition of protein with subsequent readings taken on the cumulative addition of the appropriate enzyme in the same buffer, with corrections made for dilution. Transfer efficiencies (E) were determined according to equation (1), where $F_{DA}$ and $F_D$ represent the fluorescent signal of the DAL and DOL DNA at the given wavelengths, respectively (for

example, $F_{DA}(\lambda^D_{EX}, \lambda^A_{EM})$, denotes the measured fluorescence of acceptor emission on excitation of the donor, for DAL DNA); $\varepsilon^D$ and $\varepsilon^A$ are the molar absorption coefficients of donor and acceptor at the given wavelengths; and $\varepsilon^D(490)/\varepsilon^A(560)$ and $\varepsilon^A(490)/\varepsilon^A(560)$ are determined experimentally from the absorbance spectra of DAL and the excitation spectra of singly TAMRA-AOL, respectively. Energy transfer efficiency (E) was fitted by non-linear regression in the Kaleidagraph program to equation (2), where $E_{max}$ and $E_{min}$ are the maximum and minimum energy transfer values, [S] is the substrate concentration, [P] is the protein concentration and $K_{bend}$ is the bending equilibrium dissociation constant of the protein substrate [PS] complex.

$$E = (\text{ratio})_A / \left( \frac{\varepsilon^D(490)}{\varepsilon^A(560)} \right) - \left( \frac{\varepsilon^A(490)}{\varepsilon^A(560)} \right), \tag{1}$$

Where $(\text{ratio})_A = \frac{F_{DA}\left(\lambda^D_{EX}, \lambda^A_{EM}\right) - N \cdot F_D\left(\lambda^D_{EX}, \lambda^A_{EM}\right)}{F_{DA}\left(\lambda^D_{EX}, \lambda^D_{EM}\right)}$,

And $N = F_{DA}\left(\lambda^D_{EX}, \lambda^D_{EM}\right) / F_D\left(\lambda^D_{EX}, \lambda^D_{EM}\right)$,

$$E = E_{min} + \frac{(E_{max} - E_{min})}{2[S]} \left[ ([S] + [P] + K_{bend}) - \sqrt{([S] + [P] + K_{bend})^2 - 4[S][P]} \right]$$
$$\tag{2}$$

Donor (fluorescein) was excited at 490 nm with emission sampled as the average value of the signal between 515 and 525 nm, and acceptor (TAMRA) was excited at 560 nm with emission averaged between 580 and 590 nm.

**Multiple turnover rates.** Reaction mixtures (final volume 180 μl) were prepared in 1.5 ml microcentrifuge tubes with 50 nM final substrate concentration (S5,1; S0,1-5P; S0,1-5OH; or S0,1-5FAM) and incubated at 37 °C before addition of enzyme to initiate the reaction. The final composition of each reaction mixture was 1 × RB (55 mM HEPES pH 7.5, 110 mM KCl, 8 mM MgCl$_2$, 0.1 mg ml$^{-1}$ BSA) supplemented with 1 mM DTT. Enzyme concentrations were chosen to give ∼15% cleavage after 20 min, and any data points showing greater cleavage were discarded due to effects of substrate depletion. For substrates S5,1 and S0,1-5FAM, aliquots (20 μl) of each reaction mixture were quenched into 250 mM EDTA (50 μl) at seven different time points—typically 2, 4, 6, 8, 10, 12 and 20 min—and reaction progress monitored by dHPLC analysis using a WAVE system equipped with an OligoSep cartridge (4.6 × 50 mm; ADS Biotec). The 6-FAM label was detected by fluorescence (excitation 494 nm, emission 525 nm) and product(s) separated from unreacted substrate using the following gradient: 5–30% B over 1 min; 30–55% B over 4.5 min; 55–100% B over 1.5 min; 100% B for 1.4 min; ramp back to 5% B over 0.1 min; hold at 5% B for 2.4 min, where A is 0.1% v/v MeCN, 1 mM EDTA, 2.5 mM tetrabutylammonium bromide and B is 70% v/v MeCN, 1 mM EDTA, 2.5 mM tetrabutylammonium bromide[12]. Initial rates ($v$, nM min$^{-1}$) were determined by linear regression of plots of product concentration versus time and adjusted for enzyme concentration to give normalized rates ($v/[E]$, min$^{-1}$). For analysis of exonucleolytic activity, reactions with substrates S0,1-5P and S0,1-5OH were run as above but quenched in 98% deionised formamide containing 10 mM EDTA. Time points and enzyme concentrations were selected to give 10–15% cleavage at the reaction end point ($\geq 20$ min). The quenched samples were analysed by capillary electrophoresis as detailed below, then rates determined and normalized as above.

**Analysis of reaction aliquots by capillary electrophoresis.** Capillary electrophoresis was performed with the P/ACE MDQ Plus system (Beckman Coultier) using the ssDNA 100-R Kit (AB SciEx UK Limited; #477480) according to the manufacturer's instructions. Briefly, the supplied capillary (ID 100 μm, 30 cm long; 20 cm to detection window) was loaded with the commercially supplied gel using 70 psi of pressure for 5 min. The capillary was then equilibrated between two buffer vials containing Tris-Borate-Urea buffer (AB SciEx UK Limited; #338481) at 3, 5 and 9.3 kV for 2, 2 and 10 min, respectively, with a ramp time of 0.17 min. Samples were then run using a 5 s electrokinetic injection preceded by a 1 s plug injection of deionised water, before separation over 20 min with a voltage of 9.3 kV applied between two buffer vials; runs were carried out at 50 °C with constant pressure of 40 psi maintained on both sides of the capillary. The gel was replaced every five sample runs and running buffer was replaced every 20 sample runs. Peak detection was by laser induced fluorescence (LIF) using an excitation wavelength of 488 nm and a 520 nm filter to measure the emission. The electrophoretograms were integrated to determine the concentration of product formed at each time point. Initial rates of reaction ($v$, nM min$^{-1}$) were then obtained using linear regression, and converted to the reported normalized rates ($v/[E]$, min$^{-1}$) as above.

**Single turnover rapid quench experiments.** Rapid quench experiments for determination of single turnover rate were carried out for wt hFEN1 and the mutants R104A, K132A, R103AR129A and D233N. Reactions were carried out at 37 °C using an RQF-63 device (HiTech Limited, Salisbury, UK)[12,70]. Premix stock solutions of enzyme and substrate were prepared at 2 × final concentration in reaction buffer (55 mM HEPES pH 7.5, 110 mM KCl, 8 mM MgCl$_2$, 2.5 mM DTT and 0.1 mg ml$^{-1}$ BSA) and kept on ice until use. For individual reactions, the two 80 μl sample injection loops of the instrument (lines A and B) were filled with aliquots of enzyme and substrate stock, respectively. The syringe feeding the

quench line contained 1.5 M NaOH, 20 mM EDTA. Individual reactions were carried out using a controlled time delay of between 0.0091 and 51.241 s before quenching, with final concentrations of 5 nM substrate S5,1 and either 400 nM or 1,000 nM enzyme, as indicated (Supplementary Fig. 4C,D). Quenched reaction mixtures were analysed by dHPLC as described above for multiple turnover reactions, and rates were derived from curves consisting of at least 14 individual time points. The single turnover rate of the reaction was obtained as the first-order rate constant ($k_{ST}$) derived using nonlinear least squares regression for a one- or two-phase exponential in GraphPad Prism 6.05 (GraphPad Software, Inc.). Model selection was by statistical analysis using Aikake's Information Criteria (AIC).

**Benchtop single turnover experiments.** For the remaining proteins—hFEN1 mutants R104AK132A, R103ER129E, R104EK132E, QUAD-E (R103E/R104E/R129E/K132E) and D181A—reactions to determine single turnover rates were carried out using manual sampling, as described for the multiple turnover reactions above, except using 5 nM substrate S5,1 and an enzyme concentration of either 400 nM or 1,000 nM as indicated in each case (Supplementary Fig. 4C,D). A final reaction volume of 360 μl was used, permitting sampling of 14 time points per tube, which were typically chosen to span a reaction duration of at least 20 half-times. Quenched samples were analysed by dHPLC as detailed above, then single turnover rates were derived as described for the rapid quench experiments.

**Yeast strain construction.** To construct the individual yeast mutants, the hphMX4 hygromycin resistance marker was first integrated downstream of Rad27, replacing genomic region ChrXI:224,681–224,712, in a strain containing the Ura3-(GAA)$_{100}$ cassette[49] derived from parent strain CH1585 (MATa leu2-Δ1, trp1-Δ63, ura3–52, and his3–200). The rate of (GAA)$_{100}$ expansion in this strain (designated Rad27-Hyg) was indistinguishable from the wild type strain not carrying the downstream hphMX4 cassette. Genomic DNA from Rad27-Hyg was used as a template for PCR with a ~100 bp forward primer containing the specific mutations and a reverse primer downstream of the hphMX4 cassette. These PCR products were used to transform the wt (GAA)$_{100}$ strain with selection on 200 μg ml$^{-1}$ hygromycin. Transformants were screened by PCR and/or restriction digest, and the full-length sequences of the mutated Rad27 alleles were verified by Sanger sequencing. The length of the starting (GAA)$_{100}$ tract in the mutant strains was confirmed by PCR using primers A2 (5′-CTCGATGTGCAGAACCTGAAGCTTGATCT-3′) and B2 (5′-GCTCGAGTGCAGACCTCAAATTCGATGA-3′).

**Yeast spot assay.** Fivefold serial dilutions were made on an equivalent starting number of cells for each strain. A 2.5 μl aliquot of each dilution was spotted onto YPD, YPD with 10 μg ml$^{-1}$ camptothecin, or YPD with 100 mM hydroxyurea. For UV treatment, cells spotted onto YPD were immediately irradiated using a UV Stratalinker 1,800 (Stratagene).

**Fluctuation assay and expansion rates.** At least two independent isolates of each yeast mutant were diluted from frozen stocks and grown for 40 h on solid rich growth media (YPD) supplemented with uracil. 16 individual colonies (8 per isolate) were dissolved in 200 μl of water and serially diluted. Appropriate dilutions were plated on synthetic complete media containing 0.09% 5-fluoro-orotic acid (5FOA) to select for large-scale expansion events or YPD to assess total cell number. Colonies on each plate were counted after three days of growth. For each mutant, at least 96 representative 5FOA colonies (8–12 per plate) were analysed for large-scale GAA expansion via PCR with primers A2 and B2 followed by agarose gel electrophoresis (1.5% agarose in 0.5X TBE). To determine a true expansion rate (as opposed to a gene inactivation rate), the number of 5FOA-r colonies counted per plate was adjusted by the overall percentage of GAA expansion events observed for that mutant. Expansion rates were calculated using the Ma-Sandri-Sarkar maximum likelihood estimator method with a correction for plating efficiency determined as $z-1/z \ln(z)$, where $z$ is the fraction of the culture analysed (Rosche and Foster, 2000). PCR product lengths for the calculation of GAA expansion size were determined using cubic spline interpolation on Total Lab Quant software. Kolmogorov-Smirnov comparison of expansion lengths between strains was conducted using SPSS software—non-parametric testing of independent samples. Genotype information for each strain used is shown in Supplementary Table 1.

**Extraction of Rad27 proteins and western blotting.** Wt and mutant strains in mid-log phase (OD$_{600}$ 0.6–0.8, 10 mls) were pelleted, washed with water and frozen. Pellets were resuspended in 150 μl of distilled water, mixed with an equal volume of 0.6 M NaOH with a 10 min incubation at room temperature. After low speed centrifugation (153 g) for 5 min, the supernatant was removed and each pellet resuspended in SDS sample loading buffer (60 mM Tris-HCl pH 6.8, 4% β-mercaptoethanol, 4% SDS, 0.01% bromophenol blue, 5% glycerol). The samples were boiled for 3 min, then 10 μl of each separated by 4–12% SDS–PAGE gel (Invitrogen) followed by Western blotting using anti-RAD27 goat polyclonal antibody (1:125 dilution; Santa Cruz Biotechnology, #sc-26719), donkey anti-goat IgG-HRP secondary antibody (1:2,500; Santa Cruz Biotechnology; #sc-2020) and visualized using an ECL detection kit (GE Healthcare). A nonspecific band present in all lanes was used as a loading control (Fig. 5d).

**Data availability.** Coordinates and structure factors are deposited with the Protein Data Bank (PDB) under the accession codes: 5UM9 (D86N), 5KSE (R100A), and 5K97 (D233N). The data that support the findings of this study are available from the corresponding authors on request.

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

## Acknowledgements

For financial support, we thank NCI (PO1CA92584 to J.A.T.); BBSRC (grants BB/K009079/1 and BB/M00404X/1 to J.A.G.); the Ministry of Higher Education Libya, EPSRC and the University of Sheffield (studentships to S.I.A., S.J.S. and E.J., respectively); NIH (R01GM060987 to S.M. and R01GM110387 to S.T.); and KAUST for core and CRG3 funding to S.M.H. and J.A.T. J.A.T. acknowledges support by a Robert A. Welch Chemistry Chair, the Cancer Prevention and Research Institute of Texas, and the University of Texas System Science and Technology Acquisition and Retention. The tumour mutation analysis includes data generated by the TCGA Research Network: http://cancergenome.nih.gov/. We thank the ALS, SSRL, the IDAT program, the DOE BER and the NIH project MINOS (R01GM105404) for X-ray data facilities. We thank James Holton for aiding X-ray data analysis.

## Author contributions

J.A.G., L.D.F., M.J.T., S.E.T., F.R., S.M.H., S.M.M. and J.A.T. designed the experiments. M.J.T., L.D.F., S.I.A., S.J.S., V.J.B.G., M.Z.H. and E.J. made mutant proteins and performed biochemical analyses. A.S.A., S.E.T. and J.A.T. did crystallographic analyses. J.C.K., A.J.N. and S.M.M. carried out the in vivo yeast studies. A.S.S. did western blot analysis.

## Additional information

**Competing interests:** The authors declare no competing financial interests.

DOI: 10.1038/ncomms16145    OPEN

# Corrigendum: Phosphate steering by Flap Endonuclease 1 promotes 5′-flap specificity and incision to prevent genome instability

Susan E. Tsutakawa, Mark J. Thompson, Andrew S. Arvai, Alexander J. Neil, Steven J. Shaw, Sana I. Algasaier, Jane C. Kim, L. David Finger, Emma Jardine, Victoria J.B. Gotham, Altaf H. Sarker, Mai Z. Her, Fahad Rashid, Samir M. Hamdan, Sergei M. Mirkin, Jane A. Grasby & John A. Tainer

*Nature Communications* 8:15855 doi: 10.1038/ncomms15855 (2017); Published 27 Jun 2017; Updated 7 Aug 2017

The financial support for this Article was not fully acknowledged. The Acknowledgements should have included the following:

This research used resources of the Advanced Light Source and the Stanford Synchrotron Radiation Lightsource, which are DOE Office of Science User Facilities under contract no. DE-AC02-05CH11231 and DE-AC02-76SF00515, respectively. The SSRL Structural Molecular Biology Program is supported by the DOE BER, and by the NIH (including P41GM103393). The SIBYLS beamline 12.3.1 is supported by the IDAT program from the DOE BER and the NIH project MINOS (R01GM105404).

