## [Peer Review File · Nature Communications]

Reviewers' Comments:

Reviewer #1:

Remarks to the Author:

Tainer et al., 2017, Nature Communications

Tainer et al used the combined crystallographic, biochemical and genetic analyses and showed that two dsDNA binding sites set the 5' polarity and reveal the control mechanism of the DNA phosphodiester backbone by electrostatic interactions, termed as "phosphate steering". A lot of efforts have been investigated to crystallize the three mutant FEN1 proteins with ssflap DNA strand and successfully found that the basic residues steer an inverted ss 5'-flap through a gateway over FEN1's active site and shift dsDNA for efficient catalysis. They have then made four mutations on different individual steering residues and measured kinetics with three different substrates with or without ssDNA flap strand and found that the combined mutations of these residues cause an 18,000-fold reduction in catalytic rate in vitro. In addition, the authors believe that those residues are most relevant to the triple repetitive DNA expansion in the genome so that they have established the corresponding mutations in yeast Rad27 gene to observe the expansion. They found the large-scale trinucleotide (GAA)_n repeat expansions in vivo and proposed that the failed phosphate-steering promotes an unanticipated lagging strand template-switch mechanism during replication. The authors concluded that the phosphate steering is an unappreciated FEN1 function that enforces 5'-flap specificity and catalysis, preventing genomic instability. Overall the reviewer appreciates the great efforts invested in the current manuscript to summarize all the previous findings and refine the previous data to be more precise and accurate. However, it is also concerned that the work included here does not represent a major conceptual advance in the field based on the following considerations:

1. A series of papers from the same group and others regarding the various structures and structural/functional analyses of flap endonucleases (different homologues) have identified various structural elements critical for binding and interactions with different regions of DNA flap structured substrates including the ones that interact with ssDNA flap regions and demonstrated their roles. The findings included in the current manuscript are well defined and refined in the current manuscript but not completely novel.
2. All four residues R103, R104, R129 and K132 included in the current manuscript indicated to be important for the ssDNA flap binding were originally identified to be critical to bind the ss DNA flap in the previous studies by the same group more than 10 years ago (Qiu et al., JBC, 2004, Table 1). This paper should be cited in the current manuscript.
3. The trinucleotide (GAA)_n repeat expansion assays in yeast with FEN1's catalytic mutations were done and published previously (Singh et al., JBC, 2007). This paper should be cited in the current manuscript. Similar results were obtained in the previous studies. The only difference is that different mutations and different terms (phosphate steering) were used here.
4. In Figure 7B, a series of 14 mutations identified in cancer and data based was illustrated in the structure model to illustrate the significance of the current work. To improve the quality of the current manuscript, more recent literature on these mutations should be included. For example, Liu et al., 2015, EMBO J. studied the functional deficiency of A159V in mouse model.

Reviewer #2:

Remarks to the Author:

In this manuscript, Tainer and coworkers propose new ideas pertaining to the mechanism of FEN1 action pertaining to some observations that were difficult to reconcile with previously described mechanisms. The new structures, which have a more elaborate substrate, show an inverted phosphate backbone past the scissile bond. This led the authors to propose a new threading and phosphate steering idea that helps explain the selection of substrates.

These are new and interesting ideas that are clearly important to disseminate to the community. They also appear to shed light on the mechanism of triplet expansion and instability, though these should be more clearly explained.

The experiments and the logic presented in this elegant work are sound and I have just a couple of additional minor comments.

Figure 1b is a key figure, yet it takes a lot of effort to see the difference between the left most and right most schematics until one delves further into the paper. It's hard to spot the inversion in the 5' flap (it was almost like one of those 'spot the differences' puzzles). The authors should point this out more blatantly for the reader.

There is an additional substrate structure from the 2011 Cell paper by these authors (PDB ID 3Q8M). The authors should comment on that as well and what the differences are with these new structures (general differences, not minute details).

Reviewer #3:

None

Reviewer #4:

Remarks to the Author:

Comments on: "Phosphate steering by Flap Endonuclease 1 promotes 5'-flap specificity and incision to prevent genome instability" by Tsutakawa et al.

This work addresses how the FEN1 endonuclease uniquely positions and cuts the the 5' flap of its target DNA without cutting within the flap itself. By combining three FEN1-DNA crystal structures with biochemical and genetic data they present a very clear case for how FEN1 selects for 5' flaps and precisely positions the DNA backbone to avoid unnecessary cutting within the 5' flap. They furthermore show that perturbation of this selection mechanism leads to microsatellite instability in yeast.

Altogether this is a very nice piece of work, where the crystal data is complemented by the biochemistry and genetic experiments. The manuscript is clearly written and I have no comments to the work itself.

I only have a few minor comments

1) Fig 1C & 1D: Change label 'H3TH' to 'H2TH'

2) Page 6, lines 80-91. There is no mention of Tyr40 in the description of the interaction between FEN1 and DNA substrate. Yet, from the Figure 3A it is clear it plays a crucial role in the positioning of the -1 scissile phosphate and 5' flap. It would be good if the authors would the role of Tyr40 in this section. (It is currently only mentioned three pages later (p9, line 152)).

3) Fig 3C: The depiction of the metal that is present (shown in green sphere) and the missing metal (green sphere, blue outline) is very similar. It would be more clear if the missing metal was shown only in the blue outline, a star, or something similar.

4) Fig 5 & 6: The use of the cryptic names 2E-1, 2E-2, 4A etc should be avoided. Instead, they should use full names of the mutants (i.e. R105EK130E) as in the preceding figure 4. Currently, the identity of only three of the yeast mutants are defined in the text.

5) Page 11, line 202: "classical two-metal-ion catalysis". Wei Yang has recently shown that DNA synthesis uses a three-metal-ion catalysis (PMID 22785315, 27284197, 27602203). Is there any possibility that FEN1 too might use three metals?

6) Page 33-34, lines 677-708: These two sections (Fluctuation Assay and Expansion rates) describe the same experiment. Remove one.

We thank the reviewers for their detailed comments (*blue italics*). We appreciate their support for the novelty and impact. *Reviewer 2: “ These are new and interesting ideas that are clearly important to disseminate to the community. They also appear to shed light on the mechanism of triplet expansion and instability ...”* *Reviewer 4: “By combining three FEN1-DNA crystal structures with biochemical and genetic data they present a very clear case for how FEN1 selects for 5' flaps and precisely positions the DNA backbone to avoid unnecessary cutting within the 5' flap. They furthermore show that perturbation of this selection mechanism leads to microsatellite instability in yeast.”* Reviewer 1 asks that we cite 3 papers by Shen and colleagues (which are cited in the revised manuscript) and questions the advance over these earlier papers, which we clarify below, and in the revised manuscript.

We appreciate the specific suggestions by all three reviewers to improve the paper for publication. We have therefore revised the manuscript to appropriately incorporate all the reviewers' comments, as noted below.

Reviewer #1.

Reviewer 1 asks that we cite 3 papers by Dr. Shen and colleagues and questions the conceptual advance in regard to these three papers: *“However, it is also concerned that the work included here does not represent a major conceptual advance in the field based on the following considerations.”* In fact, this reviewer misjudges the content of the prior work by Shen and colleagues compared to this manuscript: the conceptual advances can be simply summarized. Briefly, no prior structures of any members of the 5' nuclease superfamily show an inversion of the ss 5'-flap, have a substrate with the scissile phosphate coordinating the catalytic metals, or locate the attacking water. These multiple novel structural observations in our manuscript furthermore paved the way to specific insights into how incisions may be prevented in the ssDNA and how incision is precisely controlled at the flap junction. Supporting and extending these new structural results, the biochemistry in our manuscript has identified novel and catalytically critical roles for the cap residues not only in interacting with the ssDNA but also with the duplex DNA. Moreover, neither mutations of these basic residues in the yeast enzyme, nor the size of the expansions have been previously studied. Importantly, these mutations and expansions sizes are evaluated quantitatively in our manuscript, which led to our novel proposal that the lagging strand may switch templates. This is another conceptual change, as it has been thought that only the leading strand can do this.

1. *“A series of papers from the same group and others regarding the various structures and structural/functional analyses of flap endonucleases (different homologues) have identified various structural elements critical for binding and interactions with different regions of DNA flap structured substrates including the ones that interact with ssDNA flap regions and demonstrated their roles. The findings included in the current manuscript are well defined and refined in the current manuscript but not completely novel.”*

Our current manuscript provides novel information on structural features that were missing for eukaryotic flap endonucleases and their paralogs including interaction with

the ss 5'-flap and positioning of the scissile phosphate for catalysis. The ss 5'-flap is the defining structural element of the 5'-flap substrate and the presence in our current structures is a major advance. This result has enabled us to observe 1) that the 5'-flap passes through the gateway in an orientation with the phosphodiester further away from the metals than the bases, 2) that the substrate moves into the active site through a rolling mechanism with base pairing intact, and 3) the position of the attacking water. None of these findings were previously possible, as published structures did not have a similarly inverted ss 5'-flap or did not have the scissile phosphate catalytically positioned.

2. "All four residues R103, R104, R129 and K132 included in the current manuscript indicated to be important for the ssDNA flap binding were originally identified to be critical to bind the ss DNA flap in the previous studies by the same group more than 10 years ago (Qiu et al., JBC, 2004, Table 1). This paper should be cited in the current manuscript."

The referee advises that because there was an earlier paper (Qiu et al., JBC, 2004) that described a survey of charge mutations to FEN1 with semi-quantitative evaluation of their activities (ranked as – to +++ with no statistics or error bars), the combined structural, biochemical, and biological results on R103, R104, R129 and K132 in our work is not novel. In fact, this earlier work by the Shen group with us predates knowledge of the preferred human FEN1 substrate. So it did not assay the mutated proteins with the correct substrate with two duplex arms: a 5'-flap and a one nt ss 3'-flap. This coupled with the somewhat error-prone single data point approach taken to kinetic and binding assays at that time (rather than single-turnover assays or measurements in the linear part of the activity curve), actually led to misleading interpretations regarding the importance of residues noted in our manuscript. E.g. Qiu et al report R103 as wild-type and K132A to be catalytically inactive and to abolish DNA binding. In contrast, our more rigorous study shows a small (3-5 fold) impact on the rate of reaction and a negligible alteration of DNA binding ability for these residues. Yet, we agree with the prior general proposal that *"the large loop (now referred to as the helical gateway and cap) is involved in DNA interactionsand could participate in catalysis."* Building upon this and other prior studies, our work reveals the key molecular interactions by cap and gateway residues to define the mechanistic roles of these residues. This includes promoting an inverted orientation of the ss 5'-flap, duplex DNA interactions, and a critical rate-limiting catalytic step. So we now cite the noted paper and clarify the novelty.

3. The trinucleotide (GAA)_n repeat expansion assays in yeast with FEN1's catalytic mutations were done and published previously (Singh et al., JBC, 2007). This paper should be cited in the current manuscript. Similar results were obtained in the previous studies. The only difference is that different mutations and different terms (phosphate steering) were used here.

We agree with Rev. 1 and now cite the paper of Singh et al., 2007. In a nutshell, it showed that (GAA)_n and (CTG)_n repetitive flaps were poorly processed in the Rad27 knockout and Rad27-E176A mutant in vitro (not studied in our manuscript). Also, expansions and fragility of (CTG)_n repeats were elevated in those mutants in vivo. We added this reference to the three relevant statements in the MS. Notably, expansions of

(GAA)_n repeats as well as the scale of (CTG)_n repeat expansions, which prompted our novel proposal that lagging strand can participate in template switches, were not measured in the Singh paper.

4. *“In Figure 7B, a series of 14 mutations identified in cancer and data based was illustrated in the structure model to illustrate the significance of the current work. To improve the quality of the current manuscript, more recent literature on these mutations should be included. For example, Liu et al., 2015, EMBO J. studied the functional deficiency of A159V in mouse model.”*

The mutants were mapped were from the human cancer genome not mouse models. However, we have included the reference as suggested.

Reviewer #2 (Remarks to the Author):

“Figure 1b is a key figure, yet it takes a lot of effort to see the difference between the left most and right most schematics until one delves further into the paper. It’s hard to spot the inversion in the 5’ flap (it was almost like one of those ‘spot the differences’ puzzles). The authors should point this out more blatantly for the reader.”

We thank the reviewer for the suggestion. We have added text and arrows to the figure and improved the schematic.

“There is an additional substrate structure from the 2011 Cell paper by these authors (PDB ID 3Q8M). The authors should comment on that as well and what the differences are with these new structures (general differences, not minute details).”

The 3q8m structure is a mutant D181A structure with no metals. In general, it’s similar to 3q8l, with a substrate with no ss 5’-flap. Without active site metals, the DNA is repulsed away from the active site carboxylates, compared to 3q8l (0.5 Å RMSD overall). Tyr40 is in the same rotamer. We did not include discussion of this structure, as it is similar to 3q8l and it’s a nuanced difference that may confuse or distract readers from the main points.

Reviewer #4

1) *“Fig 1C & 1D: Change label ‘H3TH’ to ‘H2TH’”*

We changed the label to H2TH.

2) *“Page 6, lines 80-91. There is no mention of Tyr40 in the description of the interaction between FEN1 and DNA substrate. Yet, from the Figure 3A it is clear it plays a crucial role in the positioning of the -1 scissile phosphate and 5’ flap. It would be good if the authors would the role of Tyr40 in this section. (It is currently only mentioned three pages later (p9, line 152).”*

We agree. We have now mentioned Tyr40 earlier in the text.

3) *“Fig 3C: The depiction of the metal that is present (shown in green sphere) and the missing metal (green sphere, blue outline) is very similar. It would be more clear if the missing metal was shown only in the blue outline, a star, or something similar.”*

We have now shown the missing metal with only the blue outline.

4) *“Fig 5 & 6: The use of the cryptic names 2E-1, 2E-2, 4A etc should be avoided. Instead, they should use full names of the mutants (i.e. R105EK130E) as in the preceding figure 4. Currently, the identity of only three of the yeast mutants are defined in the text.”*

We wished to avoid confusing readers by the changed numbering between human and yeast. We now include a table that shows each group with their respective human and yeast numbering in Figure 3 to avoid confusion and provide full names of mutants as suggested.

5) *“Page 11, line 202: “classical two-metal-ion catalysis”. Wei Yang has recently shown that DNA synthesis uses a three-metal-ion catalysis (PMID 22785315, 27284197, 27602203). Is there any possibility that FEN1 too might use three metals?”*

It is indeed possible. We have now mentioned it in the text.

6) *“Page 33-34, lines 677-708: These two sections (Fluctuation Assay and Expansion rates) describe the same experiment. Remove one.”*

Thank you. We have fixed this in the revised manuscript. We are grateful to Rev. 4 for pointing out to the unnecessary duplication of the fluctuation test description in the Methods part of the MS.